# Homodimer-mediated phosphorylation of C/EBPα-p42 S16 modulates acute myeloid leukaemia differentiation through liquid-liquid phase separation

Dongmei Wang[1,2,4], Tao Sun [1,2,4], Yuan Xia[1], Zhe Zhao[1], Xue Sheng[1], Shuying Li[1], Yuechan Ma[1], Mingying Li[1], Xiuhua Su [1], Fan Zhang[3], Peng Li[1], Daoxin Ma [1,2], Jingjing Ye[1], Fei Lu[1,5] ✉ & Chunyan Ji [1,2,5] ✉

CCAAT/enhancer binding protein α (C/EBPα) regulates myeloid differentiation, and its dysregulation contributes to acute myeloid leukaemia (AML) progress. Clarifying its functional implementation mechanism is of great significance for its further clinical application. Here, we show that C/EBPα regulates AML cell differentiation through liquid-liquid phase separation (LLPS), which can be disrupted by C/EBPα-p30. Considering that C/EBPα-p30 inhibits the functions of C/EBPα through the LZ region, a small peptide TAT-LZ that could instantaneously interfere with the homodimerization of C/EBPα-p42 was constructed, and dynamic inhibition of C/EBPα phase separation was observed, demonstrating the importance of C/EBPα-p42 homodimers for its LLPS. Mechanistically, homodimerization of C/EBPα-p42 mediated its phosphorylation at the novel phosphorylation site S16, which promoted LLPS and subsequent AML cell differentiation. Finally, decreasing the endogenous C/EBPα-p30/C/EBPα-p42 ratio rescued the phase separation of C/EBPα in AML cells, which provided a new insight for the treatment of the AML.

CCAAT/enhancer-binding protein alpha (C/EBPα) is a basic-leucine zipper (bZIP) transcription factor and serves as a main regulator of myeloid differentiation and leukaemogenesis[1,2]. Two major C/EBPα protein isoforms exist: the 42 kDa full length protein (p42) and the 30 kDa truncated isoform (p30), which lacks a transactivation domain, compared with the C/EBPα-p42[3,4]. The previous studies proved that C/EBPα enhanced cell differentiation and blocked the cell cycle progression by blocking the miR-182 expression[5], activating *CSF3R*[6] or *IL6R*[7], repressing *CEBPG*[8] and E2F[9], respectively. Meanwhile, C/EBPα was necessary for long-term self-renewal and lineage initiation of

hematopoietic stem cells[10], and improved the curative effects of the LSD1 inhibitor in the treatment of AML[11].

C/EBPα mutation had been widely used in the clinical diagnosis and treatment of AML[12,13]. The majority of AML patients with C/EBPα mutation are biallelic mutations. The selective loss of C/EBPα-p42 was found in these patients because they usually have one allele carrying the N-terminal mutation and the other carrying the C-terminal mutation[14–16]. As we all know, the N-terminal mutations often expressed C/EBPα-p30 which could form C/EBPα-p30 homodimers, and the C-terminal mutations could either block dimerization of C/EBPα with

[1]Department of Hematology, Qilu Hospital of Shandong University, Jinan, Shandong, China. [2]Shandong Key Laboratory of Immunohematology, Qilu Hospital of Shandong University, Jinan, Shandong, China. [3]Department of Critical Care Medicine, Qilu Hospital of Shandong University, Jinan, Shandong, China. [4]These authors contributed equally: Dongmei Wang, Tao Sun. [5]These authors jointly supervised this work: Fei Lu, Chunyan Ji. ✉e-mail: lufeisdu2@163.com; jichunyan@sdu.edu.cn

itself or with other members of the C/EBP family. Thus, C/EBPα-p30 homodimers are the only functional C/EBPα dimers in these patients and work independently of C/EBPα-p42 by different mechanisms. For example, C/EBPα-p30 sustained leukaemic growth via the CD73/A2AR axis[17]; it alleviated immunosuppression of CD8 + T cells by inhibiting autophagy-associated secretion of IL-1β in AML[18]. Meanwhile, the previous studies also reported that C/EBPα-p30 is generally considered to be an inhibitor of C/EBPα-p42 in normal genotype cells. Mechanistically, this inhibition is mainly due to the formation of C/EBPα-p30:C/EBPα-p42 heterodimers, which, in contrast to C/EBPα-p42 homodimers, impair the transactivation- and DNA-binding capacity[4,19–23]. In summary, the functions of C/EBPα in various pathophysiological processes have been clarified. Further clarification of the specific molecular mechanism to achieve its functions is of great significance for the development of related drugs targeted at C/EBPα.

Liquid–liquid phase separation (LLPS) has been recognized as a mechanism that regulates crucial biological processes[24–26]. Many biological regulatory factors including transcription factors[27], gene promoters[28] and super-enhancers[29], have been confirmed to induce the LLPS via chemical modification[30,31], DNA[32] or RNA binding[33] and the formation of multiprotein complexes[34]. Clarifying the mechanism of phase separation of specific molecules and its regulatory effects on molecular functions is crucial for biological processes and even diseases.

Here, we show that C/EBPα promotes AML cell differentiation through LLPS, which can be disrupted by its dominant negative mutant C/EBPα-p30. Mechanistically, homodimer-mediated phosphorylation of C/EBPα S16 modulates the LLPS and thus governs AML differentiation. At last, we decreased the endogenous C/EBPα-p30/C/EBPα ratio to rescue the phase separation of C/EBPα in AML cells and significantly enhance drug efficacy. Our study clearly depicts a model in which LLPS is dependent on homodimers-mediated phosphorylation, which lays a theoretical foundation for understanding the regulatory mechanism of LLPS by an endogenous dominant negative mutant.

## Results

### C/EBPα undergoes an abnormal particle alteration during cell differentiation

The combination of Ara-C and Dox has been used as the standard first-line chemotherapy for AML patients. Inducing differentiation of AML cells is believed as an important mechanism for chemotherapeutic drugs[35–37]. In our GFP + MLL-AF9 AML mouse models (Fig. 1a–c), we observed that the number of endogenous C/EBPα particles in GFP-tagged MLL-AF9 cells was significantly increased after being treated with Ara-C and Dox (Fig. 1d and Supplementary Fig. 1a), suggesting that the C/EBPα particles may be involved in the cell differentiation induced by chemotherapeutic drugs. Furthermore, we isolated CD34+ primary AML cells from AML patients (*n* = 6) and treated them with Ara-C and Dox in vitro. As expected, the number of endogenous C/EBPα condensates was gradually increased after the addition of both drugs (Fig. 1e and Supplementary Fig. 1b). Similar results were also observed in PMA-induced differentiation of AML cell line THP-1[38] (Fig. 1f and Supplementary Fig. 1c). In addition, we also detected the endogenous C/EBPα condensates in two other differentiation systems. Briefly, we confirmed the differentiation of 3T3-L1 cells into adipocytes by Oil Red O assay, and found that endogenous C/EBPα condensates were significantly increased in adipocytes (Supplementary Fig. 1d, e). Moreover, we observed the increase of endogenous C/EBPα condensates during the differentiation of 32Dcl3 cells into granulocytes (Supplementary Fig. 1f, g). Collectively, the results indicate that endogenous C/EBPα undergoes an abnormal particle alteration when it induces cell differentiation.

### C/EBPα undergoes LLPS

LLPS is a newly recognized organization form of intracellular proteins, which mainly displays as protein particles. To verify whether the endogenous C/EBPα condensates were from LLPS, we analysed the amino acid sequence of C/EBPα with the IUPred protein structure prediction server and found that it contained disordered regions that tend to undergo LLPS (Fig. 2a). Then, PEG was added to purified EGFP-tagged C/EBPα protein (Supplementary Fig. 2a) to evaluate whether C/EBPα formed LLPS in vitro. Compared with the purified EGFP protein, the C/EBPα solution exhibited turbidity under crowding conditions (Fig. 2b). Furthermore, we observed that purified EGFP-C/EBPα protein formed micrometre-sized spheres and then could be substantially eliminated after 2 min treatment with 10% 1,6-hexanediol (1,6-Hex) prior to imaging, a compound that interferes with weak hydrophobic interactions and inhibits droplet formation (Fig. 2c). However, purified EGFP proteins had no significant change (Fig. 2c). To further illustrate the biophysical properties of the purified EGFP-C/EBPα heterotypic droplets, we operated fluorescence recovery after photobleaching (FRAP) experiment to assess the dynamics of the droplets. Notably, the droplet showed a relatively rapid fluorescence recovery (~40 s, 80% recovery) after bleaching, suggesting that the droplets were in a liquid-like state (Fig. 2d). Simultaneously, the phase-separated droplets exhibited rapid fusion observed in the fluoresce and differential interference contrast (DIC) (Fig. 2e).

To test whether C/EBPα also undergoes LLPS in cells, we overexpressed EGFP-tagged C/EBPα in HEK 293 T cells (Supplementary Fig. 2b). Considering that some transcription factors proteins, such as YAP1[39] and SEUSS[40], should be activated by LLPS to regulate gene transcription upon hyperosmotic stimulation, we used mannitol-induced osmotic stress to induce condensation of C/EBPα in HEK 293 T cells. After 10 min hyperosmotic stress stimulation with 0.1 M mannitol, we found small green granules of C/EBPα became clearer and more apparent in the nucleus of overexpressed C/EBPα-EGFP HEK 293 T (Fig. 2f, g and Supplementary Fig. 2d). In addition, the granules were also abolished after 2 min treatment with 10% 1,6-Hex (Fig. 2f), which was similar to the results in vitro. Meanwhile, the EGFP group failed to show condensates following 0.1 M mannitol stress in HEK 293T-EGFP cells (Supplementary Fig. 2c). Then, we further illustrated the dynamics of the C/EBPα droplets using FRAP assay. The analysis of FRAP showed 80% fluorescence recovery within 60 s after bleaching (Fig. 2h) and reconfirmed the LLPS of C/EBPα in HEK 293 T overexpressed EGFP-tagged C/EBPα cells. Moreover, the fusion of C/EBPα-EGFP puncta was also noticed (Fig. 2i). In summary, C/EBPα undergoes LLPS. In addition, C/EBPα is a member of the C/EBP (CCAAT/enhancer binding protein) family which also includes C/EBPβ, C/EBPδ, C/EBPε, C/EBPγ, and C/EBPζ. To probe the LLPS potential of other C/EBP family members, we expressed C/EBPδ, C/EBPε and C/EBPγ in HEK 293 T cells but did not observe puncta formation after 10 min of hyperosmotic stress (Supplementary Fig. 2e).

C/EBPα is a transcription factor. To confirm the influence of phase separation on transcription function of C/EBPα, we performed CHIP-sequencing (CHIP-seq) and RNA-seq for PMA treated THP-1 cells. The results of GSEA analysis of CHIP-seq data showed that the phase-separated C/EBPα activated some differentiation-related pathways in PMA treated THP-1 cells, including developmental growth involved in morphogenesis, myeloid cell differentiation, developmental maturation and macrophage activation. Meanwhile, GSEA analysis of RNA-seq data also showed the phase-separated of C/EBPα activated the same differentiation-related pathways (Supplementary Fig. 2f–h). In general, the LLPS of C/EBPα promotes cell differentiation by binding with the promoter of some key downstream molecules.

### C/EBPα-p30 inhibits the LLPS of C/EBPα

Clinically, C/EBPα mutations occur in approximately 10% of AML cases, and leading to expression of N-terminally truncated C/EBPα-p30 protein, which lacks the first 119 amino acids (Fig. 3a)[3,4]. To prove whether C/EBPα-p30 also undergoes LLPS, we studied the formation of C/EBPα-p30 puncta in the HEK 293 T overexpressed EGFP-tagged C/EBPα-p30.

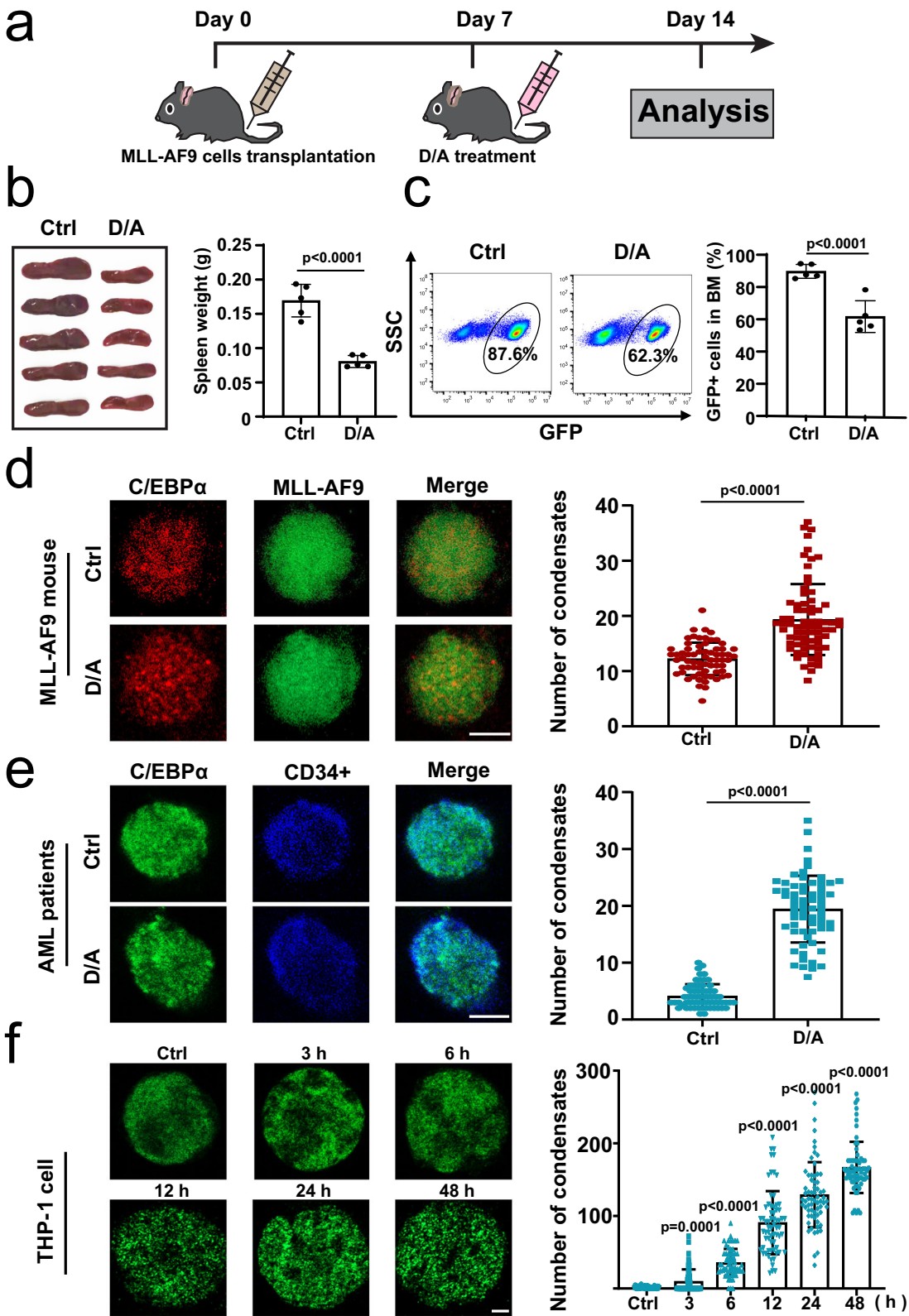

The failing to produce turbidity and diffuse appearance was observed, which suggested that C/EBPα-p30 had lost the ability to phase separation (Fig. 3b–d, Supplementary Fig. 3a, b).

C/EBPα-p30 had been proved to neutralize the transcriptional activity of C/EBPα by interfering the formation of C/EBPα-p42 homodimers[21,41]. And 36-residue leucine zipper (LZ) domain was the direct binding domain between C/EBPα-p42 homodimers or C/EBPα-

p42:C/EBPα-p30 heterodimers (Fig. 3e)[42]. To investigate whether C/EBPα-p30 binding to C/EBPα-p42 to inhibit LLPS by its LZ domain, a Flag-tagged C/EBPα-p30ΔLZ was constructed. Then, C/EBPα-p30ΔLZ was proved lost the ability to dimerize with C/EBPα in immunoprecipitation assay (Fig. 3f). Then, we performed a concentration gradient experiment in mannitol stimulated-HEK 293 T cells, including the mass ratio of C/EBPα-p30 construct to full-length C/EBPα construct as 1:1,

**Fig. 1 | C/EBPα undergoes an abnormal particle alteration during cell differentiation. a** Experimental schematic for the establishment of the Dox/Ara-C (D/A) treated GFP + MLL-AF9 mouse model. **b** Representative photographs and weights of spleens isolated from the groups of GFP + MLL-AF9 mice at 7 days after treatment ($n = 5$ mice per group). A two-tailed unpaired Student's $t$ test was used for statistical analysis and data were presented as mean values ± SEM. **c** Flow cytometric analysis of GFP + MLL-AF9 leukaemia cells in BM of the control and D/A groups ($n = 5$ mice per group). A two-tailed unpaired Student's $t$ test was used for statistical analysis and data were presented as mean values ± SEM. **d** Immunofluorescence photograph analysis of the number of endogenous C/EBPα condensates in each GFP + MLL-AF9 leukaemia cell in the control and D/A groups. Quantification of C/EBPα condensates ($n = 3$ biologically independent experiments). A two-tailed unpaired Student's $t$ test was used for statistical analysis, and data were presented as mean values ± SEM. **e** Immunofluorescence photograph analysis of the number of endogenous C/EBPα condensates in each CD34+ primary AML cell treated with the control or D/A (Ara-C:50 nM; Dox:0.2 μg/mL) for 24 h. Quantification of C/EBPα condensates ($n = 6$ AML patients). A two-tailed unpaired Student's $t$ test was used for statistical analysis, and data were presented as mean values ± SEM. **f** Immunofluorescence photograph analysis of the number of endogenous C/EBPα condensates in each THP-1 cell stimulated with 100 ng/mL PMA for 0 h, 3 h, 6 h, 12 h, 24 h and 48 h. Quantification of C/EBPα condensates ($n = 3$ biologically independent experiments). A two-tailed unpaired Student's $t$ test was used for statistical analysis, and data were presented as mean values ± SEM. The experiments in (**d**, **f**) were observed at least 20 random fields in each experiment. For (**e**), at least 10 random fields were observed in each experiment. For (**d**, **e**), scar bars, 5 μm. For (**f**), scar bar, 2 μm. Source data are provided as a Source Data file.

3:1, 9:1 and 18:1. The results showed that the number of condensates in 1:1 group was obviously more than that in 3:1 group. Meanwhile, the condensates of mCherry-tagged C/EBPα-p30 were observed in these two groups, which might because of the formation of C/EBPα-p42: C/EBPα-p30 heterodimer (Supplementary Fig. 3d, e). Furthermore, no C/EBPα condensate was observed in 9:1 or 18:1 group (Supplementary Fig. 3d, e). Similar results have also been obtained in experiment by using purified mCherry-tagged C/EBPα-p30 and C/EBPα-EGFP proteins (Supplementary Fig. 3f). Collectively, as the mass ratio of C/EBPα-p30 to C/EBPα increases, the condensates gradually decreased until they disappeared. The mass ratio of C/EBPα-p30 construct to full-length C/EBPα construct 9:1 is the lowest mass ratio for C/EBPα-p30 to suppress C/EBPα LLPS, and was selected in subsequent experiments. More interestingly, the phase separation was significantly inhibited in HEK 293 T cells that were co-transfected with the C/EBPα-EGFP and mCherry-C/EBPα-p30, while mCherry-C/EBPα-p30ΔLZ had no influence on the LLPS of C/EBPα-EGFP (Fig. 3g and Supplementary Fig. 3c). It is worth noting that in the presence of C/EBPα, C/EBPα-p30ΔLZ shows a certain degree of aggregation. But in the absence of C/EBPα, C/EBPα-p30ΔLZ particles disappear (Supplementary Fig. 3g). These results suggested that C/EBPα-p30ΔLZ cannot undergo LLPS. Meanwhile, considering that C/EBPα-p30ΔLZ and C/EBPα had no direct interaction and some reports mentioned that they could bind to the same gene promoters[43,44], we speculated that the LLPS of C/EBPα changed the distribution and functional states of its binding chromosomes, which led to the redistribution of C/EBPα-p30ΔLZ which had similar chromosome binding ability as wild type C/EBPα. Overall, these data support that C/EBPα-p30 does not undergo phase separation and may inhibit the LLPS of C/EBPα by interrupting the formation of C/EBPα-p42 homodimers.

## The LZ fragment inhibits the LLPS of C/EBPα−42 by disrupting the formation of its homodimers

To further determine C/EBPα-p30 inhibit LLPS of C/EBPα-p42 by disrupting the formation of its homodimers, the HEK 293 T cells were transfected with mCherry-tagged C/EBPα-p30 or LZ fragment together with C/EBPα-p42 (Supplementary Fig. 4a). By using NATIVE PAGE, C/EBPα-p42 monomer, C/EBPα-p30 monomer, C/EBPα-p42 homodimer, C/EBPα-p30 homodimer and C/EBPα-p42:C/EBPα-p30 heterodimer were observed (Fig. 4a). These results proved that C/EBPα-p30 or LZ formed heterodimers with C/EBPα-p42 through LZ domain. Since C/EBPα-p30 could dimerize with itself, we also tested whether its homodimers undergo LLPS. As expected, we didn't observe the LLPS of C/EBPα-p30 homodimers (Supplementary Fig. 4b). More importantly, the interfering effects of C/EBPα-p30 or LZ for the formation of C/EBPα-p42 homodimer were proved (Fig. 4a). In addition, we also verified that the binding ability with C/EBPα became stronger as the amount of LZ plasmids increased from 0.5 μg to 4 μg, indicating that the LZ fragment could interact with C/EBPα (Fig. 4b). Importantly, in HEK 293 T cells transiently transfected with C/EBPα-EGFP, no

condensate in the mCherry-tagged LZ group was observed compared with the control group after mannitol treatment (Fig. 4c), suggesting that LZ fragment inhibits the LLPS of C/EBPα.

In order to observe the effect of LZ fragment on the C/EBPα LLPS instantaneously, we designed a small peptide which could efficiently transport into the cells in several minutes. Briefly, the peptide was named as TAT-LZ and consisted of TAT and the LZ fragment (36 amino acids) conjugated with 5-TAMRA at the C-terminus (Fig. 4d). The TAT-LZ peptide gradually crossed the membrane into HEK 293 T cells within 5 min, and the massive intracellular abundance was observed at 10 min (Supplementary Fig. 4c). Then, we found a significant increase in droplet numbers within 10 min in the no treat group (mannitol alone) of HEK 293 T overexpressing C/EBPα-EGFP (Fig. 4e, f and Supplementary Fig. 4d). Meanwhile, the number of droplets increased within 3 min and decreased at 5-10 min when TAT-LZ and mannitol were added simultaneously at 0 min, compared with that in no treat group (Fig. 4e, f and Supplementary Fig. 4d). Moreover, when TAT-LZ was added 5 min before mannitol, the number of droplets were further reduced (Fig. 4e, f and Supplementary Fig. 4d). And the addition of TAT-LZ 10 min before mannitol completely blocked the formation of LLPS condensates (Fig. 4e, f and Supplementary Fig. 4d). Similarly, the number of endogenous C/EBPα particles in THP-1 cells stably expressing the LZ fragment was greatly reduced after PMA induction, compared with that in control cells (Fig. 4g, h, Supplementary Fig. 4e). Given that C/EBPα is a key regulatory molecule for monocyte and macrophage differentiation[45,46], we used qRT-PCR and flow cytometry to study the differentiation of THP-1 cells and found that the LZ fragment significantly reduced the expression of *CD11b* or *CD68* and decreased the percentage of CD11b or CD68 positive cells (Fig. 4i, j and Supplementary Fig. 4f). Taken together, our data indicate that the LZ fragment inhibits the LLPS of C/EBPα-p42 and the differentiation of AML cells by disrupting the formation of its homodimers.

## Phosphorylation of C/EBPα-p42 at S16 stabilizes its LLPS through homodimerization

To investigate how the homodimers enhance the LLPS of C/EBPα-p42, we employed the mass spectrometry analysis to study several kinds of chemical modification and found that phosphorylation of C/EBPα at Ser16 was significantly increased when the HEK 293 T cells overexpressing C/EBPα-EGFP being treated with mannitol, which may be relevant to the liquid droplet formation by phase separation (Fig. 5a). Consequently, the rabbit-derived anti-C/EBPα p-S16 antibody was designed and produced. Meanwhile, ELISA results showed that p-S16 antibody had up to five-fold stronger affinity for C/EBPα p-S16 peptide than C/EBPα S16 peptide (Supplementary Fig. 5a). Subsequent western blot analysis reconfirmed that the relative p-S16 level got stronger with time under mannitol treatment (Supplementary Fig. 5b). Similarly, the endogenous p-S16/S16 level in THP-1 cells also increased after being treated with PMA for 0, 3 h, 6 h, 12 h and 24 h (Fig. 5b). Interestingly, we found that C/EBPα-S16 levels decrease over the 24 h period of PMA

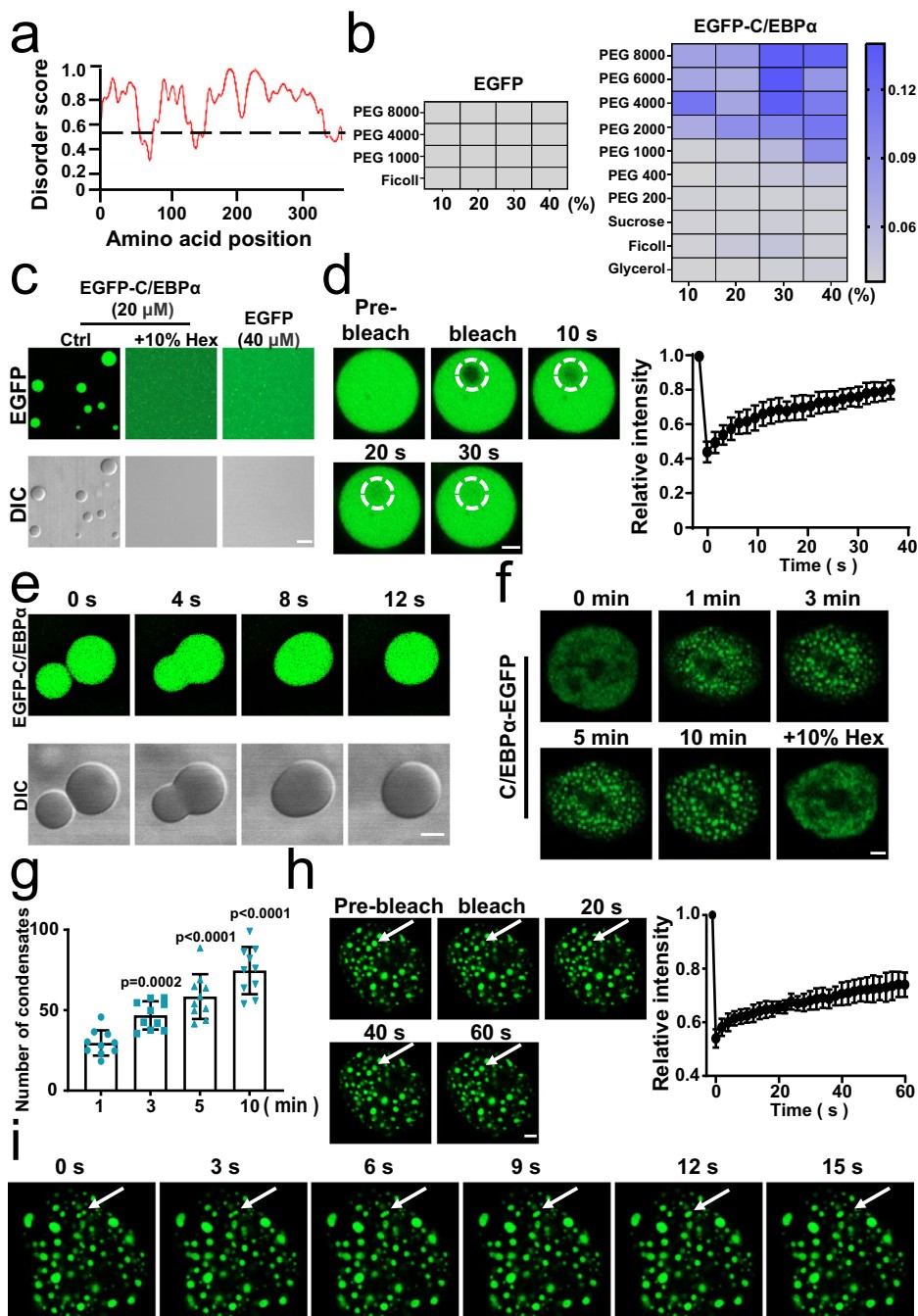

**Fig. 2 | C/EBPα undergoes LLPS. a** Analysis of disordered regions in C/EBPα. IUPred was used to analyse the regions of C/EBPα; Regions with a score of greater than 0.5 (above the dotted line) were considered disordered. **b** Turbidity of purified EGFP-C/EBPα protein (20 μM) increased after treatment with crowding agents, while EGFP protein did not. **c** EGFP-C/EBPα (20 μM) droplets were formed after treatment with 250 mM NaCl and 10% PEG 8000 and then disappeared after 2 min addition of 10% 1,6-Hex, while EGFP did not exhibit these properties. **d** Representative FRAP images of EGFP-C/EBPα droplets. The dotted circle shows the bleached area. The FRAP curve of EGFP-C/EBPα droplets was analysed with different droplets (*n* = 8). Data are presented as mean values ± SEM. **e** Fusion of EGFP-C/EBPα droplets was imaged in the fluorescence and DIC channels (*n* = 3 biologically independent experiments). **f** C/EBPα formed puncta in HEK 293 T

living cells transfected with C/EBPα-EGFP after 0.1 M mannitol treatment and then disappeared after 2 min the addition of 10% 1,6-Hex (*n* = 3 biologically independent experiments). **g** Quantification of C/EBPα puncta in Fig. 2f (*n* = 10 cells). A two-tailed unpaired Student's *t* test was used for statistical analysis, and data were presented as mean values ± SEM. **h** Representative FRAP images of C/EBPα puncta in HEK 293 T living cells transfected with C/EBPα-EGFP after 0.1 M mannitol treatment. The white arrow indicated bleached area. The FRAP curve of C/EBPα-EGFP droplets was analysed with different droplets (*n* = 8 cells). Data are presented as mean values ± SEM. **i** Fusion of C/EBPα-EGFP droplets in the HEK 293 T living cells transfected with C/EBPα-EGFP after 0.1 M mannitol treatment was imaged (*n* = 3 biologically independent experiments). For (**c**), scar bar, 5 μm. For (**d–f**, **h**, **i**), scar bars, 2 μm. Source data are provided as a Source Data file.

treatment, which is consistent with Korbinian Brand's previous work[47]. In conclusion, we confirmed that phosphorylation of C/EBPα at Ser16 was involved in the LLPS of C/EBPα. Furthermore, overexpression of either C/EBPα-p30 or LZ decreased the phosphorylation of C/EBPα

S16, while overexpression of C/EBPα-p30ΔLZ had no such effect in HEK 293 T cells co-transfected with C/EBPα-EGFP (Supplementary Fig. 5c). Likewise, in THP-1 cells transiently transformed with C/EBPα-Flag, the relative C/EBPα p-S16 level was also decreased after co-transfection

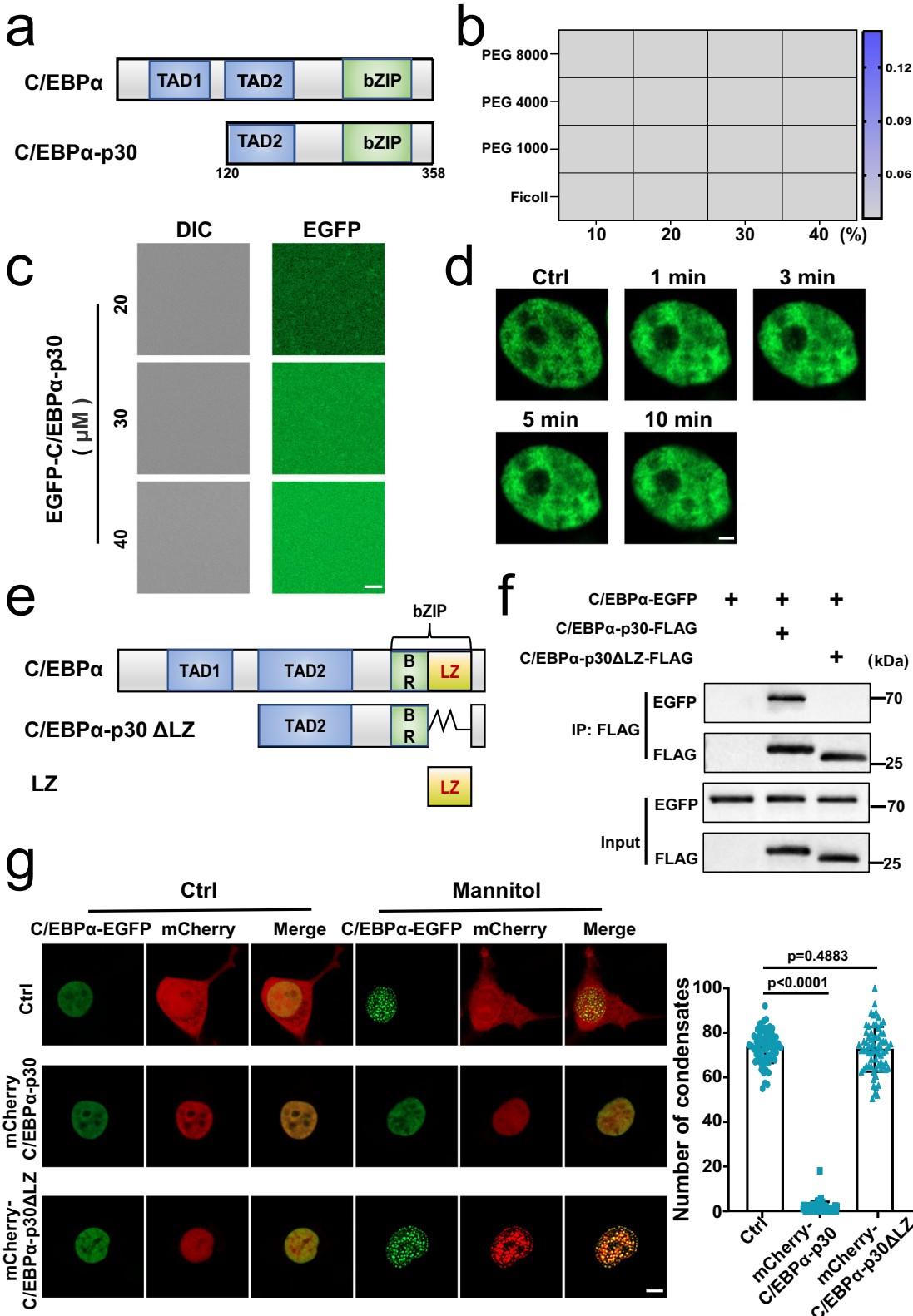

with either C/EBPα-p30 or LZ but not after co-transfection with C/EBPα-p30ΔLZ (Fig. 5c). Collectively, these data indicated that the phosphorylation of C/EBPα-p42 at S16 may regulate LLPS through homodimerization.

To further confirm whether C/EBPα S16 phosphorylation regulates LLPS in cells, we constructed the recombinant plasmids S16E (activated phosphorylation) and S16A (inactivated phosphorylation)

with EGFP tags. The results of natural gel electrophoresis showed that C/EBPα-S16E promoted the formation of C/EBPα-p42 homodimer, while C/EBPα-S16A inhibited the formation of C/EBPα-p42 homodimer in C/EBPα-S16E or C/EBPα-S16A together with C/EBPα overexpressed HEK 293 T cells (Supplementary Fig. 5d). As expected, the C/EBPα S16A mutant could not form phase-separated condensates; however, cells transfected with the S16E and wild-type C/EBPα contained obvious

**Fig. 3 | C/EBPα-p30 inhibits the LLPS of C/EBPα. a** Domain structure analysis of C/EBPα and C/EBPα-p30. **b** Purified EGFP-C/EBPα-p30 protein (20 μM) showed no turbidity. **c** EGFP-C/EBPα-p30 protein (20-40 μM) droplets did not form after the addition of 250 mM NaCl and 10% PEG 8000 (*n* = 3 biologically independent experiments). **d** No C/EBPα-p30-EGFP puncta was observed after 0.1 M mannitol treatment in HEK 293 T living cells transfected with C/EBPα-p30-EGFP (*n* = 3 biologically independent experiments). **e** The diagram of the interaction between C/EBPα and C/EBPα-p30 through the LZ domain. **f** FLAG-tagged C/EBPα-p30 or C/EBPα-p30ΔLZ was constructed into HEK 293 T cells together with C/EBPα-EGFP. The interaction was analysed by co-immunoprecipitation with anti-FLAG and anti-

GFP antibodies followed by western blotting (WB) (*n* = 3 biologically independent experiments). **g** Live-cell images indicated that the number of C/EBPα-EGFP puncta in mannitol-treated HEK 293 T cells over-expressed with C/EBPα-EGFP to mCherry-tagged C/EBPα-p30 or C/EBPα-p30ΔLZ at a mass ratio of of 1:9. Quantification of C/EBPα-EGFP puncta (*n* = 3 biologically independent experiments). A two-tailed unpaired Student's *t* test was used for statistical analysis, and data were presented as mean values ± SEM. The experiments in (**g**) were observed at least 20 random fields in each experiment. For (**c**, **g**), scar bars, 5 μm. For (**d**), scar bar, 2 μm. Source data are provided as a Source Data file.

---

granules, and the number of S16E condensates was higher than that of wild-type condensates in both HEK 293 T cells and THP-1 cells (Supplementary Fig. 5e, f, Fig. 5d, e). These results suggested that phosphorylation of C/EBPα at S16 promoted the C/EBPα LLPS. Moreover, compared with that in WT group, the enhanced phosphorylation of C/EBPα S16 strengthened the expression levels of *CD11b* and *CD68* mRNA and the percentages of CD11b+ and CD68+ cells, whereas low differentiation state were found in THP-1 cells expressing the S16A mutant (Fig. 5f, g and Supplementary Fig. 5g). Moreover, C/EBPα-S16E and C/EBPα-S16A showed similar abilities in 3T3-L1 and 32Dcl3 differentiation models (Supplementary Fig. 5h, i). Then, we studied their function in MLL-AF9 AML mouse. Compared with the control group, we found that C/EBPα-p30 significantly promoted the progression of AML, while both C/EBPα S16E and C/EBPα inhibited the progression of AML and the inhibitory effect of C/EBPα S16E was stronger than that of the WT group (Fig. 5h, i). In summary, phosphorylation of C/EBPα at S16 stabilizes LLPS through homodimerization to regulate AML differentiation.

### Decreasing the endogenous C/EBPα-p30/C/EBPα ratio rescues the function of C/EBPα promoting AML differentiation through its LLPS

To further confirm that homodimer-mediated phosphorylation of C/EBPα-p42 modulates AML cells differentiation by LLPS, we regulated the endogenous C/EBPα-p30/C/EBPα-p42 ratio. Previous studies have demonstrated that rapamycin regulates the ratio of C/EBPα-p30 to C/EBPα by decreasing the C/EBPα-p30 protein level[48,49]. So, we utilized rapamycin for the next experiments. After being proved that rapamycin significantly attenuated the C/EBPα-p30 level at 6 h, 12 h, 24 h and 48 h but did not cause any change in the C/EBPα level in THP-1 cells (Fig. 6a), 24 h was chose as the treatment duration. Confocal imaging showed that rapamycin in combination with PMA increased the number of endogenous C/EBPα particles more than PMA alone and that this effect could be counteracted by LZ fragment (Fig. 6b and Supplementary Fig. 6a), suggesting that decreased the C/EBPα-p30/C/EBPα ratio could regulate the LLPS of C/EBPα. In addition, the study on the expression levels of *CD11b* and *CD68* mRNA indicated that rapamycin enhanced the promoting effects of PMA on THP-1 cell differentiation, but this effect disappeared when LZ fragment was added (Fig. 6c). Moreover, the analysis of flow cytometry also reconfirmed this consequence (Fig. 6d and Supplementary Fig. 6b). Considering that rapamycin regulates AML through a variety of pathways, such as inhibiting AML cell viability via circ_0094100/miR-217/ATP1B1 axis[50] or enhancing the anti-leukaemia effect as a specific inhibitor of mTOR[51], we overexpressed C/EBPα-p30 in THP-1 cells treated with PMA and rapamycin. The results of qRT-PCR and flow cytometry analysis showed that C/EBPα-p30 reversed the differentiation promoting effect of rapamycin (Fig. 6c, d and Supplementary Fig. 6b), which suggesting that rapamycin promoted differentiation mainly by down-regulation of C/EBPα-p30. In summary, decreasing the endogenous C/EBPα-p30/C/EBPα-p42 ratio enhances the AML cell differentiation by promoting the LLPS of C/EBPα.

Furthermore, a novel mScarlet and GFP tagged MLL-AF9 AML mouse model was established (Fig. 6e). Firstly, we proved that

rapamycin reduced the relative levels of C/EBPα-p30 and C/EBPα-p42 in MLL-AF9 mice treated with Ara-C and Dox (Supplementary Fig. 6c). Then, consistent with the in vitro findings, rapamycin in combination with Ara-C and Dox could significantly reduce the spleen weight and the percentage of mScarlet+ GFP + MLL-AF9 cells in bone marrow, as compared with the results of the two chemotherapeutic agents. Importantly, LZ fragment eliminated the effects of rapamycin (Fig. 6f, g). Moreover, rapamycin in combination with Ara-C and Dox significantly strengthened the C/EBPα particle numbers in MLL-AF9 cells, while the effect of rapamycin could also be abolished by LZ (Fig. 6h and Supplementary Fig. 6d). Then, to prove our conclusion, we established AML xenograft mouse models by injecting NSG mice with mScarlet tagged LZ-overexpressed THP-1 cells. Expectedly, the results were identical with that in MLL-AF9 AML mouse model (Supplementary Fig. 6e–j). In summary, decreasing the endogenous C/EBPα-p30/C/EBPα ratio rescues the function of C/EBPα promoting AML differentiation through its LLPS.

## Discussion

C/EBPα is a key transcription factor regulating cell differentiation[1,52]. Our results show that C/EBPα formed dozens of nuclear puncta through LLPS, which contributes to AML cell differentiation. C/EBPα-p30 and the small peptide TAT-LZ were verified to abolish the LLPS of C/EBPα-p42 by disrupting its homodimerization. Furthermore, homodimer-mediated phosphorylation of C/EBPα-p42 at S16 site enhances C/EBPα phase condensation. Although C/EBPα-p30ΔLZ does not undergo LLPS, it prevents condensates when C/EBPα exists. Given that there is no direct binding between C/EBPα-p30ΔLZ and C/EBPα, and previous studies have reported that they could bind to the same DNA[43,44], we hypothesized that LLPS of C/EBPα changed the distribution and functional state of its binding chromosomes, resulting in the redistribution of C/EBPα-p30ΔLZ with similar chromosome binding ability to wild-type C/EBPα. However, the detailed mechanisms of this result still remain unclear and need further study. Our work highlights a mechanism that homodimer-mediated phosphorylation of C/EBPα-p42 modulates AML cell differentiation by LLPS, which lays a theoretical foundation for understanding the regulatory mechanism of LLPS by a natural endogenous dominant-negative mutant (Fig. 7).

C/EBPα mediates the expression of various downstream molecules by forming homodimers[16,53]. However, the mechanism by which C/EBPα achieves the precise adjustment of complex functions remains unclear. We found that purified C/EBPα protein undergoes LLPS and that the phosphorylation of S16 in C/EBPα, which is dependent on C/EBPα-p42 homodimerization, stabilizes phase separation. Current studies have shown that TAD1 of C/EBPα regulates transcription activation by interacting with components of RNA polymerase II pre-initiation complex, including TBP and transcription factor IIB[54]. TAD1 also involved in the C/EBPα-mediated E2F repression and the following cell cycle operation[9,55,56]. In our study, considering that phosphorylation of C/EBPα-p42 at S16 site was essential for the LLPS, we hypothesized that S16 site in TAD1 is essential for the structural remodelling in C/EBPα-p42:C/EBPα-p30 heterodimers and has a potential significance in TAD1 functional implementation. Understanding the molecular arrangement of the C/EBPα-p42 homodimers and the

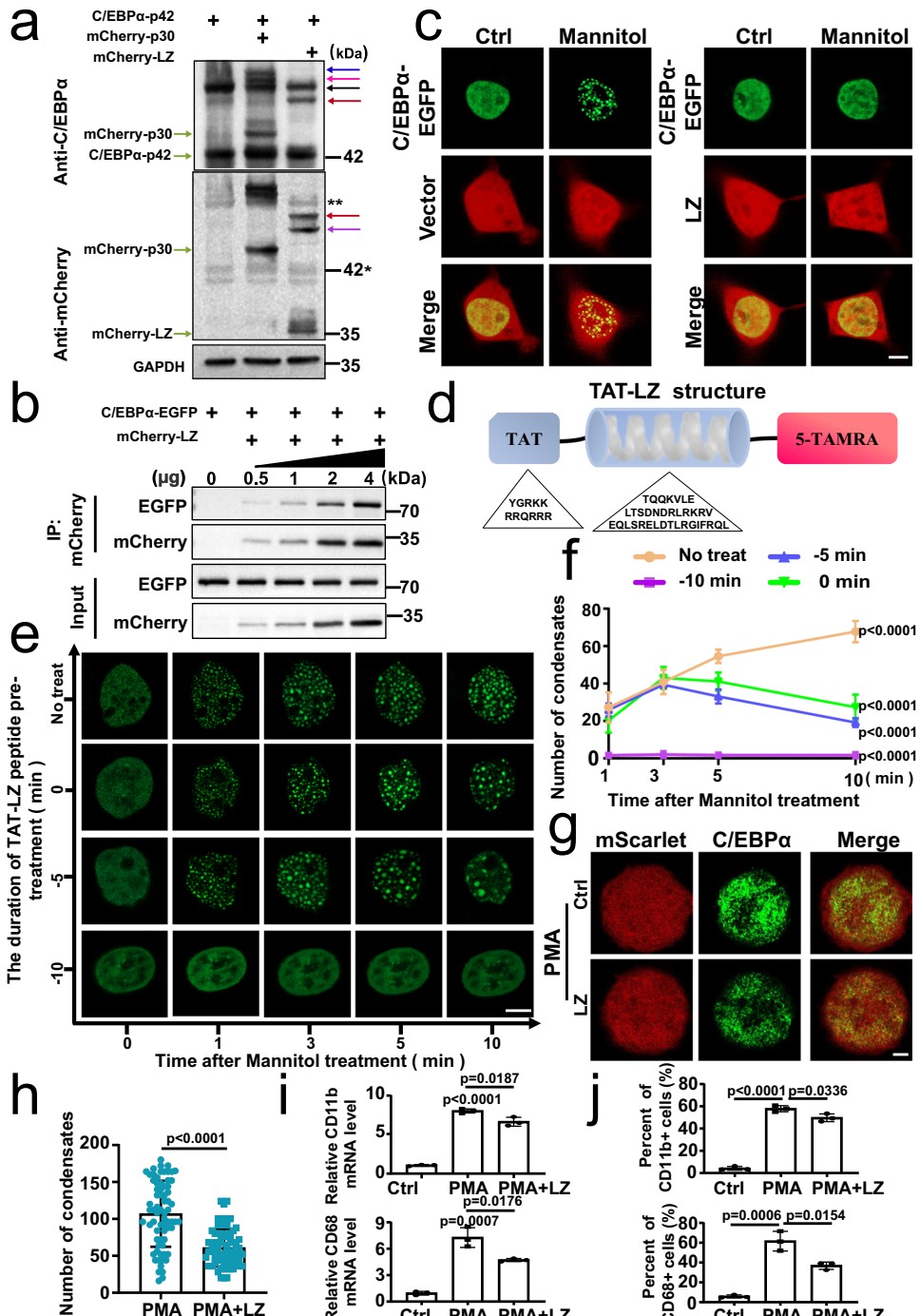

**Fig. 4 | The LZ fragment inhibits the LLPS of C/EBPα-p42 by disrupting the formation of its homodimers. a** Native gel electrophoresis show the the existence of dimer (*n* = 3 biologically independent experiments). The 0.8 μg of mCherry-tagged C/EBPα-p30 or LZ plasmids was constructed into HEK 293 T cells together with 1.2 μg of C/EBPα plasmids. Blue, pink, Black, red and purple arrows indicated the mCherry-C/EBPα-p30 homodimers, mCherry-C/EBPα-p30:C/EBPα-p42 heterodimers, C/EBPα-p42 homodimers, mCherry-LZ:C/EBPα-p42 heterodimers and mCherry-LZ homodimer respectively. "**" and "*" indicated the residual C/EBPα-p42 homodimers and C/EBPα-p42 after anti-stripping treatment; **b** mCherry-tagged LZ and C/EBPα-EGFP plasmids were co-constructed into HEK 293 T cells. The interaction was analysed by co-immunoprecipitation with anti-mCherry and anti-GFP antibodies (*n* = 3 biologically independent experiments). **c** The number of C/EBPα-EGFP puncta in mannitol-treated HEK 293 T cells transfected with C/EBPα-EGFP to mCherry-vector or LZ at a mass ratio of 1:9 (*n* = 3 biologically independent experiments). **d** Diagram of the small peptide TAT-LZ structure. **e** Changes in C/

EBPα-EGFP droplets after TAT-LZ was introduced into mannitol-treated HEK 293T-C/EBPα-EGFP living cells at −10 min, 0 min and 5 min. "x-axis" and "y-axis" indicated the time after mannitol treatment and the duration of TAT-LZ peptide pre-treatment respectively. f, Quantification of C/EBPα-EGFP puncta in Fig. 4e (*n* = 8 cells). **g** The endogenous C/EBPα condensates in THP-1 cells transduced with Ctrl or LZ under the 100 ng/mL PMA for 24 h. **h** Quantification of C/EBPα condensates for Fig. 4g (*n* = 3 biologically independent experiments). **i** The expression of *CD11b* and *CD68* in THP-1 cells transduced with Ctrl or LZ lentivirus under the 100 ng/mL PMA for 48 h (*n* = 3 biologically independent experiments). **j** The protein level of the CD11b and CD68 in THP-1 cells transduced with Ctrl or LZ lentivirus under the 100 ng/mL PMA for 24 h (*n* = 3 biologically independent experiments). For (**f**, **h**–**j**), a two-tailed unpaired Student's *t* test was used for statistical analysis, and data are presented as mean values ± SEM. For (**c**, **e**), scar bars, 5 μm. For (**g**), scar bars, 2 μm. Source data are provided as a Source Data file.

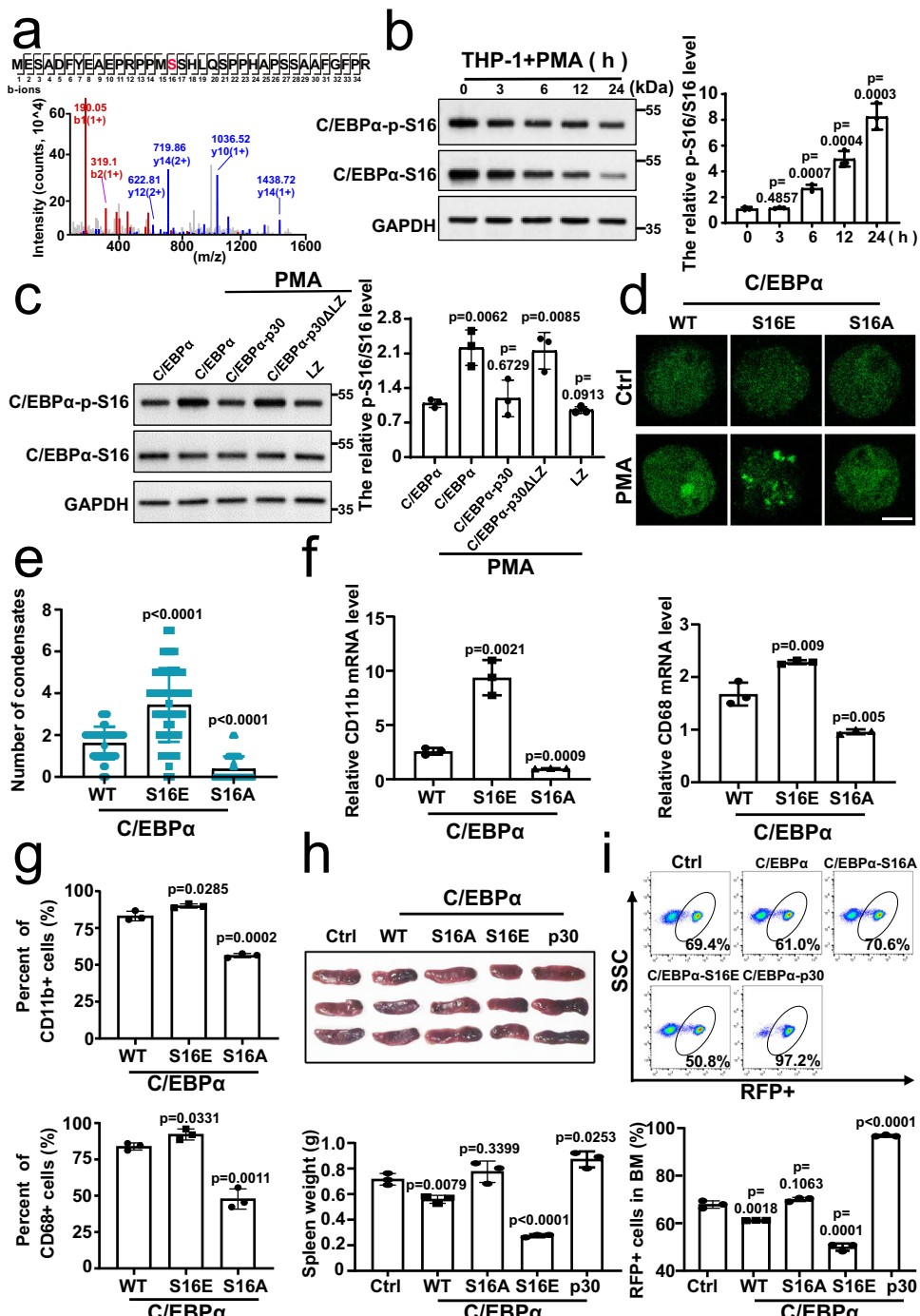

**Fig. 5 | Phosphorylation of C/EBPα at S16 stabilizes its LLPS through homo-dimerization. a** Mass spectrometry analysis indicated the C/EBPα p-S16 in HEK 293T-C/EBPα-EGFP under 0.1 M mannitol treatment for 10 min. **b** Western blot showed the level of endogenous C/EBPα p-S16 in THP-1 cells treated with 100 ng/mL PMA for 0–24 h. The samples derive from the same experiment and that gels were processed in parallel. GAPDH was run on the same blot. Quantification of the relative p-S16 level (*n* = 3 biologically independent experiments). **c** Western blot analysis showed the level of C/EBPα p-S16 in THP-1 cells transfected with p30, p30-ΔLZ, and LZ plasmids together with C/EBPα-Flag at a mass ratio of 9:1 after 100 ng/mL PMA treatment for 24 h. The samples derive from the same experiment and that gels were processed in parallel. GAPDH was run on the same blot. Quantification of the relative p-S16 level (*n* = 3 biologically independent experiments). **d** Representative images of THP-1 living cells transfected with C/EBPα-S16E-EGFP, C/EBPα-EGFP or C/EBPα-S16A-EGFP with 100 ng/mL PMA. **e** Quantification of C/EBPα condensates for 4d (*n* = 3 biologically independent experiments); At least

20 random fields were observed in each experiment. **f** The expression of the *CD11b* and *CD68* in THP-1 cells transfected with C/EBPα-S16E-EGFP, C/EBPα-EGFP or C/EBPα-S16A-EGFP with 100 ng/mL PMA (*n* = 3 biologically independent experiments). **g** The percent of CD11b+ and CD68+ in 100 ng/mL PMA-treated THP-1 cells transfected with C/EBPα-S16E-EGFP, C/EBPα-EGFP or C/EBPα-S16A-EGFP (*n* = 3 biologically independent experiments). **h** The RFP tagged control, C/EBPα-EGFP, C/EBPα-S16E, C/EBPα-S16A, C/EBPα-p30 lentiviruses were transferred into MLL-AF9 cells respectively and then injected into C57/BL6 mice. Representative photographs and weights of spleens isolated from RFP + GFP + MLL-AF9 mice in the five groups on day 10 (*n* = 3 per group). **i** Flow cytometric analysis of RFP+ leukaemia cells in BM of the five groups. Quantification of RFP+ leukaemia cells (*n* = 3 per group). For (**b**, **c**, **e**–**i**), a two-tailed unpaired Student's *t* test was used for statistical analysis, and data are presented as mean values ± SEM. For (**e**), scar bars, 5 μm. Source data are provided as a Source Data file.

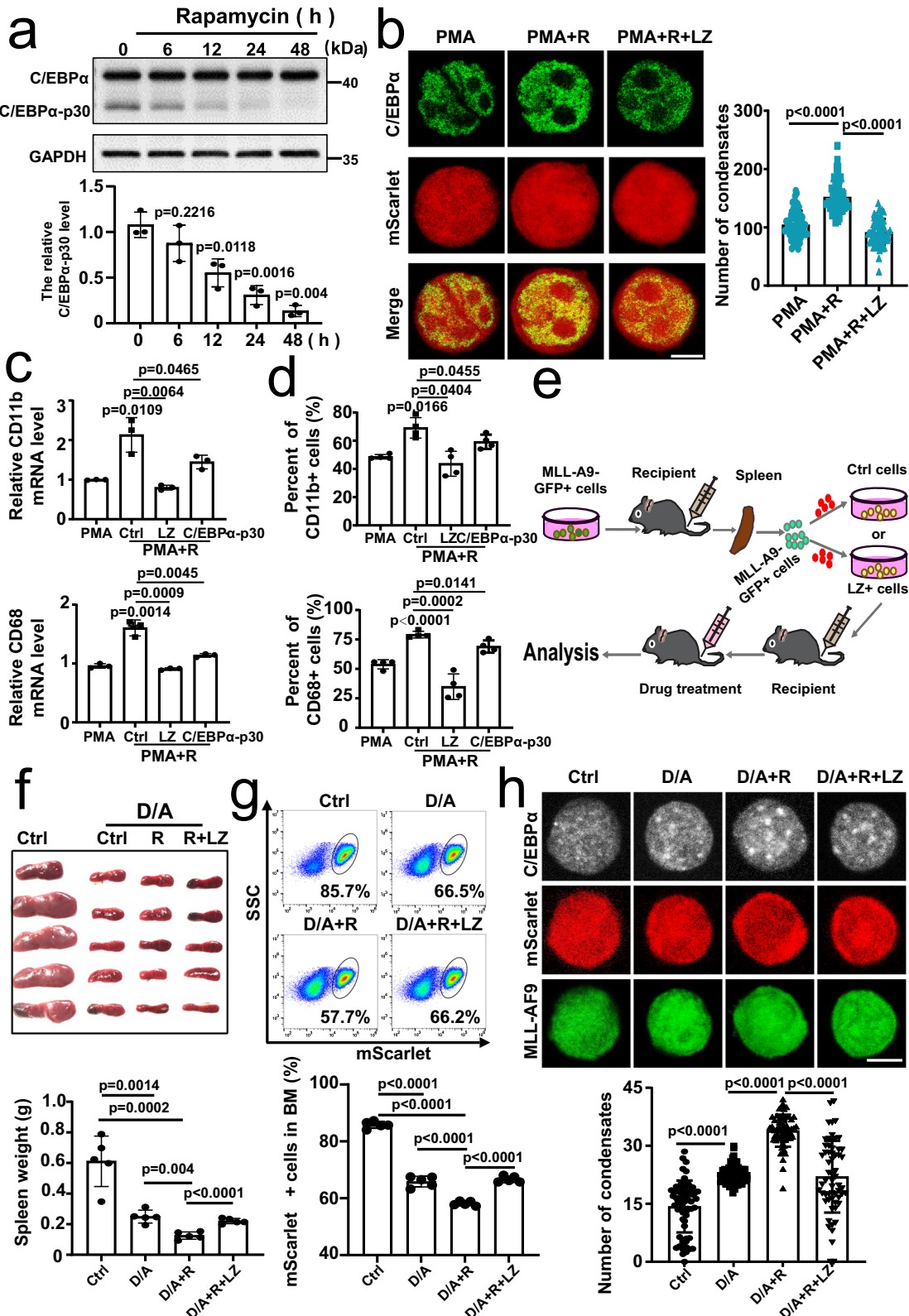

specific roles of TAD1 when phase-separated C/EBPα-p42 droplets formed would be a crucial area for future study.

As an intracellular phenomenon based on the physical mechanisms, LLPS is crucial for biological processes. The abundance of regulatory mechanisms has revealed the importance of LLPS in the cells; for example, the m6A binding proteins (YTHDF1, YTHDF2 and YTHDF3) undergo LLPS which can be markedly enhanced by the presence of mRNAs that contain multiple m[6]A residues[30]. Our present study revealed a dynamic mechanism for protein LLPS in cells: LLPS of wild-type proteins was regulated via endogenous dominant negative mutant mediated phosphorylation at specific sites. In addition, transient intervention with LLPS by the small peptides developed in this study achieved transient regulation of specific molecules in cells, which provides important direction for LLPS research.

**Fig. 6 | Decreasing the endogenous C/EBPα-p30/C/EBPα ratio rescues the function of C/EBPα through its LLPS. a** Western blot analysis showed the ratio of endogenous C/EBPα-p30 to C/EBPα in THP-1 cells treated with different concentrations of rapamycin. Relative C/EBPα-p30/C/EBPα levels (*n* = 3 biologically independent experiments). **b** mScarlet tagged control and LZ lentivirus were transduced into THP-1 cells. Immunofluorescence photograph analysed the number of endogenous C/EBPα condensates in THP-1-Ctrl or THP-1-LZ cells treated with PMA and/or PMA + RAPA (PMA + R) for 24 h. Quantification of the number of C/EBPα condensates (*n* = 3 biologically independent experiments). It was observed at least 20 random fields in each experiment. **c** mScarlet tagged control, LZ or C/EBPα-p30 lentivirus were transduced into THP-1 cells. qRT-PCR indicated the expression of CD11b and CD68 in THP-1-Ctrl, THP-1-LZ or THP-1-C/EBPα-p30 cells treated with PMA and/or PMA + RAPA (PMA + R) for 24 h (*n* = 3 biologically independent experiments). **d** Flow cytometry was performed to measure the level of the CD11b and CD68 in THP-1-Ctrl, THP-1-LZ or THP-1-C/EBPα-p30 cells treated with PMA and/or PMA + RAPA (PMA + R) for 24 h (*n* = 3 biologically independent experiments). **e** Experimental schematic for the establishment of the mScarlet+ GFP + MLL-AF9 mouse model. **f** Representative photographs and weights of spleens isolated from mScarlet+ GFP + MLL-AF9 mice in the four groups on day 14 (*n* = 5 per group). **g** Flow cytometric analysis of mScarlet+ leukaemia cells in BM of the four groups. Quantification of mScarlet+ leukaemia cells (*n* = 5 per group). h, Immunofluorescence photograph analysis of the number of endogenous C/EBPα condensates in the control, D/A, D/A + R and D/A + R + LZ groups (*n* = 5 per group). Quantification of endogenous C/EBPα condensates in each cell. For (**b**, **h**), at least 20 random fields were observed in each experiment. For (**a**–**d**, **f**–**h**, **e**–**i**), a two-tailed unpaired Student's *t* test was used for statistical analysis, and data are presented as mean values ± SEM. For (**b**, **h**), scar bars, 5 μm. Source data are provided as a Source Data file.

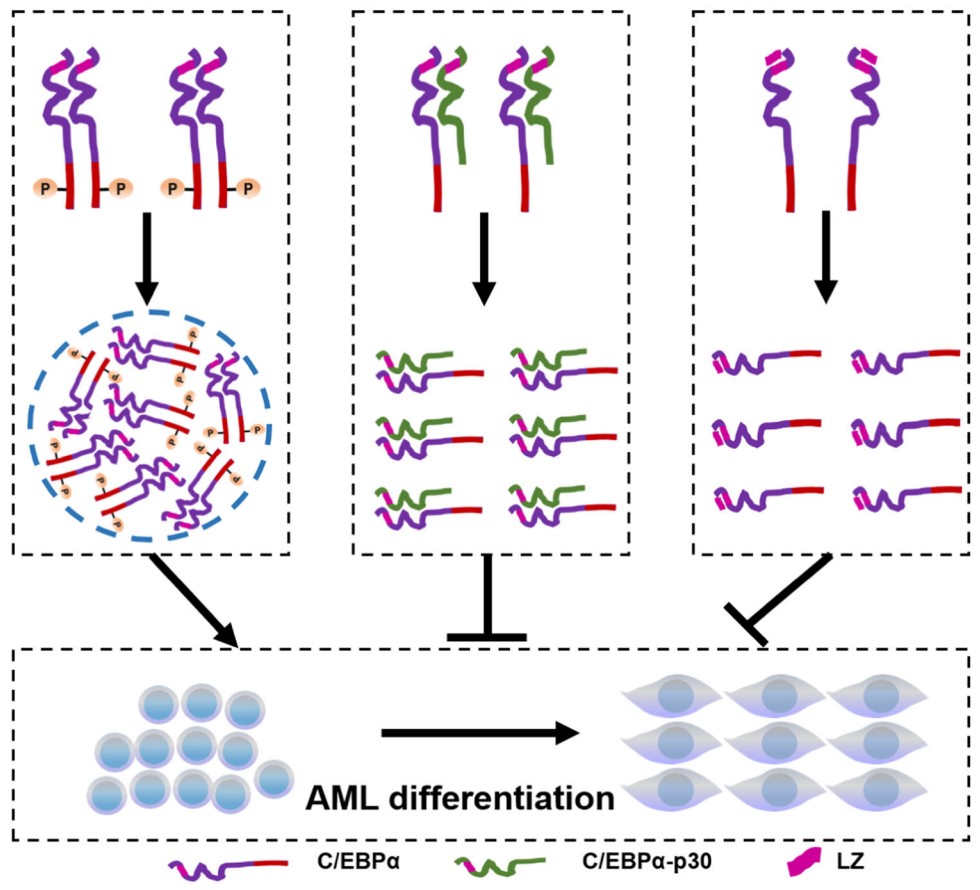

**Fig. 7 | Model for homodimer-mediated phosphorylation of C/EBPα-p42 modulating AML cell differentiation by LLPS.** Homodimer-mediated phosphorylation of C/EBPα-p42 enhances AML cell differentiation through LLPS. However, C/EBPα-p30 and LZ inhibit AML differentiation by destroying the dimerization of C/EBPα-p42 and eliminating its LLPS.

AML is characterized by a differentiation arrest and an uncontrolled proliferation of malignant blasts. Despite advances in our understanding of disease mechanisms, such as the FLT3 inhibitor gilteritinib[57,58], IDH1/2 inhibitors ivosidenib[59] and enasidenib[60] and BCL2 inhibitor venetoclax[61,62], offering considerable effect for AML, the outcome of most AML patients remains poor. Thus, novel therapeutic strategies are needed. Now, the active restoration of tumour suppressor genes is also a new direction for the development of targeted drugs to increase the therapeutic effect[63]. Here, we showed that decreasing the endogenous C/EBPα-p30/C/EBPα-p42 ratio promoted the phase separation and restored appropriate function of C/EBPα-p42 in cells. Meanwhile, our results from CHIP-seq and RNA-seq suggest that phase separation by C/EBPα promoted differentiation of AML cells through activated many differentiation-related pathways, which suggest that the LLPS of C/EBPα promotes cell differentiation by binding with the promoter of some key downstream molecules. The results reveal a new mechanism of C/EBPα regulating AML cell differentiation and provide new insights for the treatment of the AML.

In summary, we confirmed that C/EBPα enhances AML cell differentiation through LLPS. The formation of C/EBPα-p42 homodimers and phosphorylation of C/EBPα at S16 are critical for its LLPS. Strikingly, decreasing the endogenous C/EBPα-p30/C/EBPα ratio had a potential supplementary therapeutic effect through its LLPS, providing a novel therapeutic approach for AML.

## Methods

### Plasmids, antibodies, reagents

The following plasmids were designed by WZ Bioscience Inc: C-terminal 6×His tagged pET28a-EGFP-C/EBPα and pET28a-EGFP-C/EBPα-p30; pET28a-mCherry-C/EBPα-p30; C/EBPα-EGFP, C/EBPα-p30-EGFP, C/EBPα-S16E-EGFP and C/EBPα-S16A-EGFP; C/EBPα-p30-Flag, C/EBPα-p30ΔLZ-Flag, C/EBPα-Flag, LZ-Flag; mCherry-C/EBPα-p30, mCherry-C/EBPα-p30ΔLZ, mCherry-LZ. The control and LZ over-expression lentivirus were constructed using pHS-AVC (pLV-hef1a-mScarlet-P2A-Puro-WPRE-CMV-MCS-3×flag) vector, which was performed by Beijing SyngenTech Co., Ltd. The following antibodies and reagents were purchased from the indicated companies: anti-C/EBPα (Cell Signalling Technology, 2295; For IF, 1:50; For Western Blot, 1:1000); anti-mCherry (Invitrogen, M11217; 1:1000); anti-EGFP (Santa Cruz, sc-9996; 1:2000); anti-CD11b (101226, BioLegend; 1:20); anti-CD68 (333805 and 333807, BioLegend; 1:20); anti-Flag (Proteintech, 66008-4-Ig; 1:1000); anti-C/EBPα(Cell Signaling Technology, #8178; For Chip-Seq, 1:100); anti-GAPDH (Abways, AB0038; 1:5000); Goat anti-Mouse IgG antibody (Abways, AB0102, 1:5000); 1,6-Hex (Sigma); PMA (Sigma); rapamycin (Sigma); CD34 MicroBeads (Miltenyi); Ara-C (MCE); Dox (MCE); M-PER™ Mammalian Protein Extraction Reagent (Thermo Scientific); EVO M-MLV Premix (AG); SYBR® Green Premix Pro Taq HS qPCR Kit (AG); Ficoll, Mannitol, Sucrose, Milk, Glycerol, PEG 200, PEG 400, PEG 1000, PEG 2000, PEG 4000, PEG 6000 and PEG 8000 were purchased from Sangon Biotech; In addition, the anti-C/EBPα S16 and anti-p-C/EBPα S16 antibodies from rabbits were designed and produced by Bioresearch (1:1000).

### Cell culture and transfection, isolation of CD34+ primary AML cells

HEK 293 T cells were cultured in DMEM supplemented with 10% FBS and 1% penicillin/streptomycin (P/S) at 37°C, 5% $CO_2$. The 3T3-L1 cells were grown in the DMEM containing 10% Foetal calf serum and 1% P/S. 32Dcl3 cells were cultured in RPMI 1640 supplemented with 15% FBS, 1% P/S and 1 ng/ml IL-3. THP-1 cells were maintained in complete RPMI 1640 medium. THP-1 cells overexpressing control and LZ lentivirus were maintained in complete RPMI 1640 medium supplemented with 1 μg/mL puromycin. HEK 293 T and THP-1 were purchased from the Institute of Haematology and Blood Diseases Hospital, Chinese Academy of Medical Sciences and Peking Union Medical College, Tianjin, China; 32Dcl3 were sourced from Meisen CTCC, Zhejiang, China. 3T3-L1 were purchased from National collection of Authenticated Cell Cultures. Cells were transfected with the indicated plasmids using Lipofectamine 2000 and cultured for another 48 h for western blotting, qRT-PCR or live-cell imaging. Patients with AML provided informed consent at Qilu Hospital of Shandong University, and the study was approved by the Human Ethics Committee of the hospital. BMMNCs were isolated using Ficoll according to the manufacturer's protocol, followed by CD34 MicroBeads for classification of CD34+ cells.

### Differentiation of 3T3-L1 cells to adipocytes

According to previous studies[9,64], cells were fed with DEME medium containing 0.5 mM IBMX, 1 μM dexamethasone and 10 μg/ml insulin for 2 days, and then replaced with medium supplemented with only insulin after 2 days. Cells were changed to normal medium every 2 days, and harvested at eighth day for analysis of adipocytes. Oil Red O (G1262; Solarbio) were used to evaluate the adipogenesis according to the manufacturer's instructions. Photographs were taken by microscope (Invitrogen Evos FL Auto 2; USA). The quantitative oil red O staining was performed as described[65].

### Granulocytic differentiation of 32Dcl3 cells

According to previous study[66], 32Dcl3 cells with a concentration of $2 \times 10^5$ cells /ml were collected, washed twice with PBS to remove IL-3,

and then resuspended in the induction medium containing complete RPMI 1640 medium supplemented with 25 ng/ml of G-CSF (#78012_C, Stem Cell). Under the condition of the above, 32Dcl3 cells were harvested at sixth day. On the sixth day, cells were collected, and morphological evaluation was carried out using Wright-Giemsa staining solution (G1020, Solarbio) according to the manufacturer's protocol. The morphology of cells was observed by optical microscope (Nikon Eclipse Ni; Japan).

### Immunostaining

For immunostaining, $5 \times 10^5$ THP-1 cells, CD34+ primary AML cells or THP-1 cells with stable expression of control or LZ were plated and subjected to different treatments for various times at 37 °C. Then, the cells were washed with PBS, fixed with 4% formaldehyde at room temperature for 15 min and then blocked with buffer (TBS-Tx supplemented with 2% BSA and 0.1% Triton X-100) at room temperature for 30 min. Next, the cells were incubated with indicated antibodies at 4 °C overnight. The following day, the cells were incubated with secondary antibodies for 1 h before adding the antifade mounting medium. Images were acquired under a ZEISS LSM 900 confocal microscope. Primary AML cells were from 3 male and 3 female patients.

People voluntarily participated in the study by signing informed consent form. Although there are differences in age or gene mutation, it has not affected the conclusion in this study. The study was approved by the Human Ethics Committee of the Shandong University (SDULCLL2020-1-14).

### Chromatin immunoprecipitation and RNA-seq

$1 \times 10^7$ THP-1 or PMA-treated THP-1 cells were fixed in 1% formaldehyde at room temperature for 10 min, quenched with 125 mM glycine and harvested. The harvested cells were sent to Active Motif China (Shanghai, China) for ChIP-Seq. Active Motif prepared chromatin, performed ChIP reactions, generated the libraries and performed basic data analysis. In brief, chromatin was isolated by adding lysis buffer and fragmented by sonication. DNA was sheared to an average length of 200-500 bp with EpiShear probe sonicator (Active Motif, 53051). The same amount of Drosophila chromatin (Active Motif, 53083) was incubated with sheared chromatins from THP-1 cells and PMA-treated PMA cells, respectively for ChIP normalization. Genomic DNA (Input) was prepared by de-crosslinking at 65 °C for 4 h followed by RNase A and Proteinase K digestion. The Input DNA was purified with PCR Purification Kit (Qiagen, 28004) and quantified (Onedrop, Wins). Fragmented chromatin mix (for IP) was incubated with 5 µl of anti-C/EBPα (Cell Signaling Technology, 8178) and 1 µg of Drosophila H2A.v antibody (Active Motif, 61686) at 4 °C for overnight. 25 µl of rProtein G Magarose Beads (Smart-Lifesciences, SM004005) were added to the samples and incubated at 4 °C for 2 h. Complexes were washed, eluted from the beads with SDS buffer, then de-crosslinked at 65 °C for overnight. The de-crosslinked chromatin was subjected to RNase A and proteinase K treatment. ChIP DNA was purified with PCR Purification Kit (Qiagen, 28004) and quantified (Thermo Fisher, Qubit). Illumina sequencing libraries were prepared from the ChIP-DNA and input DNAs with VAHTS Universal DNA Library Prep Kit for Illumina V4. Raw reads were filtered to obtain high-quality clean reads by removing sequencing adapters, short reads (length <35 bp) and low-quality reads using trim-galore (v0.6.4). Then FastQC (v0.11.9) and Multiqc (v1.8) is used to ensure high reads quality. The clean reads were mapped to the human genome (assembly human genome hg38, primary genome) using the Burrow-Wheeler Aligner (BWA v0.7.17) software. The clean reads were mapped to the Drosophila genome (assembly Drosophila genome dm6, spike-in genome) using the Burrow-Wheeler Aligner (BWA v0.7.17) software. PCR duplicates were removed using Picard (v2.22.2-0). Scale factors were calculated by formula (number of the clean reads mapped to spike-in genome/the maximum of numbers of the clean reads mapped to spike-in genome). Subset of the clean reads

were determined by randomly downsampling using bamtools (V2.5.1), the numbers of reads to keep were calculated by formula (number of the clean reads mapped to primary genome). Peak detection was performed using the MACS2 (v2.2.6) peak finding algorithm with 0.05 set as the q-value cutoff. Annotation of peak sites to gene features was performed using the homer annotatePeaks.pl (v4.10). Pearson correlation coefficient between biological replicates was calculated using deeptools (v3.4.3) with default parameters. The resulting histograms were stored in bigWig files using deeptools (v3.4.3).

RNA-seq assay was performed on THP-1 and PMA-treated THP-1 by LC SCIENCES (Hangzhou, China). Total RNA was extracted using Trizol reagent (thermofisher, 15596018) following the manufacturer's procedure. After total RNA was extracted, mRNA was purified from total RNA (5 µg) using Dynabeads Oligo (dT) (Thermo Fisher, CA, USA). Next, the mRNA was cut into short fragments and then the cleaved RNA fragments were reverse transcribed to generate cDNA. Finally, we conducted the 2×150 bp paired-end sequencing (PE150) on an Illumina Novaseq™ 6000. The expression analysis of genes was performed by DESeq2 software.

## Protein purification

The C-terminal 6×His tagged pET28a-EGFP-C/EBPα and pET28a-EGFP-C/EBPα-p30 plasmids were overexpressed in *Escherichia coli* BL21 (DE3) cells. The cells were induced with 0.5 mM IPTG after the OD 600 reached to 0.6 and then incubated at 18°C for 12 h. The *E.coli* cells were collected, resuspended and lysed in 300 mM NaCl, 10 mM Tris and 20 mM imidazole at pH 8.0. The 6×His tagged proteins were purified with Ni-NTA Agarose (QIAGEN, 30210) and eluted in 300 mM NaCl, 10 mM Tris and 500 mM imidazole at pH 8.0. Finally, the soluble proteins were further concentrated using an Amicon Ultra-15 filter (Millipore, UFC903096). All purified proteins were quantified with a photometer (Denovix, DS-11) and stored at −80°C for the following experiments.

## In vitro droplet formation assay

This assay was carried out with the purified EGFP-C/EBPα protein incubated in 10% PEG 8000 and 250 mM NaCl at room temperature for 2 min. The protein mixture was added to 20 mm glass dishes and images were acquired with a ZEISS confocal microscope (LSM 900). Phase contrast and DIC images were acquired for each condition.

## FRAP assay

For the in vitro FRAP assay, EGFP-C/EBPα droplets were transferred to 20 mm glass dishes at room temperature. The droplets were photobleached at 100% laser power using a 488-nm laser, and images were acquired every 2 s. When the fluorescence intensities were ~40% of the prebleached values, fluorescence recovery was monitored using ZEN software, and the values were normalized to the prebleached intensities of the fluorescent C/EBPα droplets.

For the FRAP assay in live cells, HEK 293 T cells were transfected with C/EBPα-EGFP for 48 h and were then treated with 0.1 M mannitol. FRAP was monitored in C/EBPα condensates using a 488-nm laser at 100% power until the fluorescence intensities were 50% of the prebleached values. Similar to the in vitro FRAP assay, time-lapse images were acquired using a ZEISS confocal microscope (LSM 900) at intervals of 2 s, and the fluorescence intensities were normalized to the prebleached values. FRAP data were analysed for eight biological replicates per experiment.

## Co-IP and western blot analysis

For Co-IP, cells were lysed with M-PER™ Mammalian Protein Extraction Reagent (Thermo Scientific, 78501) supplemented with 1×protease and phosphatase inhibitor cocktail for 20 min at 4 °C. And then the lysate was incubated with magnetic beads that had bound the target antibody 4 °C overnight. On the second day, the protein-magnetic bead complex was boiled in 2×SDS loading buffer at 95 °C for 5 min.

The experimental samples were loaded into an SDS–PAGE gel, and proteins were subsequently transferred from the gel to a PVDF membrane. The PVDF membrane was blocked with 5% milk dissolved in TBST at room temperature and incubated with primary and HRP-conjugated secondary antibodies. The results were imaged using Chemiluminescent Imaging System (champchemi). Protein levels were quantified with Image J software.

## Native gel electrophoresis

To confirm the existence of protein dimer, we performed the Native Page. Protein was harvested from HEK 293 T which overexpressed mCherry-tagged C/EBPα-p30 or LZ together with C/EBPα. And the protein was extracted using nondenature Lysis Buffer (C510013, Sangon Biotech) without SDS. Then, the 8% Native pages were performed using Native PAGE Preparation kit (C631101, Sangon Biotech) according to the manufacturer's instructions. The lysate was loaded into a Native-PAGE gel and placed in the Tris-Glycine Native PAGE Running Buffer PH 8.8 (C506035, Sangon Biotech). Proteins were transferred to a PVDF membrane, and western analysis was performed.

## Design of the small peptide

The small peptide was synthesized by the GL Biochem (Shanghai) Ltd. The detailed sequence is as follows: TAT (YGRKKRRQRRR) + GG + LZ domain (TQQKVLELTSDNDRLRKRVEQLSRELDTLRGIFRQL) + 5-TAMRA.

## qRT-PCR

RNA isolation was carried out according to the manufacturer's protocol. RNA was reverse-transcribed using EVO M-MLV Premix. Complementary DNA was analysed using the SYBR® Green Premix Pro Taq HS qPCR Kit and RT-PCR was carried out on PCR amplifier (Roche) with GAPDH as the control. The following primers were used:

*GAPDH, forward:* GCACCGTCAAGGCTGAGAAC;
*reverse:* TGGTGAAGACGCCAGTGGA;
*CD11b, forward:* CAACGGAGCCCGAAAGAATG;
*reverse:* CTCCCACCCCAATGACGTAG;
*CD68, forward:* CTCTCCGAGTCCTAGCCCAA;
*reverse:* GCTTCCCTGGACCTTGGTTTT;

## Flow cytometry

For cell surface staining, cell suspensions were incubated with the specific labelled antibodies CD11b and CD68 at 4 °C for 30 min. The flow cytometric analyses were performed in Galias (Beckman Counter), and all data were analyzed using FlowJo 10.0.7.

## Live-cell imaging

Cells plated in glass dishes and then subjected to different treatments. Raw images were acquired with a ZEISS confocal microscope (LSM 900) with a 63×oil objective using a 488-nm or 561-nm laser. In detail, droplets were imaged and analysed using ZEN software. All images were acquired with the same parameters, and the droplets were counted using Image J. During the experiment, the cells were incubated at 37 °C in 5% CO$_2$.

## Mouse model

C57/BL6 mice aged 6-8 weeks and female NOD/SCID mice aged 3 to 4 weeks were purchased from Charles River and were maintained in the laboratory animal centre of Shandong University. Mice were maintained on a 12 h dark/light cycle at ambient temperature (72 ± 2 F) with controlled humidity (~45%). Animal protocols were approved by the Animal Ethics Committee of Shandong University (SDULCLL2020-2-12). All mice were randomly divided into the indicated groups.

For MLL-AF9 model, some C57/BL6 mice were injected with $1 \times 10^6$ MLL-AF9 cells or MLL-AF9 cells stably expressing mScarlet tagged control or LZ lentivirus after one week adaption. And then, mice were injected intravenously with 50 mg/kg Ara-C daily for 5 days and with

1.5 mg/kg Dox for the first three days of the treatment period ('5 + 3' regimen) with or without rapamycin for 5 days (1 mg/kg) after model establishment. Finally, the mice were killed and the parameters were measured in each group. Other C57/BL6 mice were injected with $1 \times 10^6$ MLL-AF9 cells or MLL-AF9 cells stably expressing RFP tagged control or C/EBPα (WT), C/EBPα-p30, C/EBPα-S16E or C/EBPα-S16A, the mice were executed after 7 days and their related parameters were observed.

For AML xenograft mouse models, male NOD/SCID mice were injected with mScarlet tagged $1 \times 10^7$ THP-1-control or THP-1-LZ cells. And then mice were injected with Ara-C and Dox with or without rapamycin as in MLL-AF9 mouse model. The weight of spleen and the percentage of positive cells in bone marrow were collected after 40 days.

### Image J analysis

A set of fluorescence images from the experiment were obtained using the same microscope settings (such as resolution, optical zoom, gain, laser intensity and scan speed) to ensure consistency between the sample and the experiment. The original picture was imported into the Image J and converted into 8 bit, and then adjusted the parameters to accurately identify particles. In order to accurately count the number of particles, we set different thresholds according to different experimental materials. For example, the threshold is set to 50–255 grey level for phase-separated particles produced by overexpressed-C/EBPα in HEK 293 T and THP-1 cells. For endogenous C/EBPα particles, set the thresholds of MLL-AF9 cells, AML primary cells, 3T3-L1 cells and 32Dcl3 as 90-255, 75-255, 85-255 and 60-255, which effectively eliminated the background signals in the analysis. The numbers of puncta of each cell within these parameters were evaluated.

### Statistical analysis

All experiments were repeated at least three times on independent days to ensure data quality and reproducibility. For the immunostaining and droplets, at least 20 images were acquired for each independent experiment. Comparisons between two conditions were performed using *Student's t-test* with GraphPad Prism 9. The data are presented as the mean ± SEM. Generally, statistical significance is shown as follows: $p < 0.05$, $p < 0.01$, $p < 0.001$, $p < 0.0001$, and statistically no significance is $p > 0.05$.

### Reporting summary

Further information on research design is available in the Nature Portfolio Reporting Summary linked to this article.

## Data availability

The raw sequencing data of CHIP-Seq and RNA-Seq generated from this study have been deposited to the GEO database under the accession codes GSE243422, GSE243423 and GSE243424. Source data are provided with this paper.

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

## Acknowledgements

We would like to acknowledge the Shandong Provincial Key Laboratory of Immunohematology (Qilu Hospital of Shandong University, Shandong Province, China) for providing the space and equipment for conducting the experiments. This work was supported by grants from the Distinguished Taishan Scholars in Climbing Plan (tspd20210321, C.Y.J), Taishan Scholars-Distinguished Experts (tstp20230653, J.J.Y), the Young Taishan Scholars (tsqn201812132, T.S), the National Natural Science Foundation of China (82070160, C.Y.J; 81873425, T.S; 82000165, D.M.W; 82170182; J.J.Y), the Major Research Plan of the National Natural Science Foundation of China (91942306, C.Y.J), the 68th China postdoctoral Science Foundation (2020M682171, D.M.W), the Key Program of Natural Science Foundation of Shandong Province (ZR2020KH016, C.Y.J; ZR2021MH302, F.L), the Fundamental Research Funds for the Central Universities (2022JC012, C.Y.J), Funded by ECCM Program of Clinical Research Centre of Shandong University (2021SDUCRCB008, T.S), the Independently Cultivate Innovative Teams of Jinan, Shandong Province (2021GXRC050, C.Y.J), the Multidisciplinary Research and

Innovation Team of Young Scholars of Shandong University (2020QNQT007.T.S).

## Author contributions

D.M.W. and T.S. designed the research. D.M.W., T.S., Y.X., Z.Z., X.S., S.Y.L., Y.C.M., M.Y.L., X.H.S. and F.Z. performed experiments. D.M.W., T.S., P.L., D.X.M., J.J.Y. and F.L. wrote the manuscript. D.M.W., T.S., J.J.Y., F.L. and C.Y.J. provided supervision.

## Competing interests

The authors declare no competing interests.
