## [Peer Review File · Nature Communications]

Homodimer-mediated phosphorylation of C/EBP α -p42 S16 modulates acute myeloid leukaemia differentiation through liquid-liquid phase separationREVIEWER COMMENTS

Reviewer #1 (Remarks to the Author):

In the present manuscript, Wang et al demonstrate that CEBPA appear to undergo liquid-liquid phase separation (LLPS). Moreover, they claim that this is driven by formation of CEBPA-p42 homodimers, that the process is dependent on phosphorylation at S16 and that it can be inhibited by expression of CEBPA-p30.

While the demonstration of CEBPA-p42 undergoing LLPS is certainly novel the manuscript, as it stands, has a number of serious flaws and also lack demonstration of the functional relevance of their findings

Major concerns

1. The authors seem to ignore a significant amount of recent literature on the function of CEBPA yielding a non-contemporary presentation of the field. Statements such as "...mechanism by which CEBPA functions is still unclear..." and that CEBPA-p30 should be "dominant negative" and that "CEBPA-p30 acts mainly by inhibiting the formation of CEBPA homodimers" are simply wrong.

2. Also the authors should use CEBPA-p42 homodimers (instead of CEBPA homodimers) as CEBPA-p30 can dimerize with CEBPA-p42 which could, in principle" be termed CEBPA homodimers.

3. The authors seem to think that role of CEBPA-p30 in CEBPA mutant AML is to prevent the formation of CEBPA-p42 homodimers. This is not correct as the CEBPA mutant AML is bi-allelic mutated and are either homozygous for the CEBPA-p30 allele or carries on CEBPA-p30 allele and an allele blocking CEBPA (any) dimerization. In essence this result in a situation where CEBPA-p30 homodimers are the only functional CEBPA dimers in these patients. The authors should modify the manuscript accordingly.

4. In order to promote LLPS the authors need to add various stressors such as mannitol, chemotherapy PEG and others. Does that mean that CEBPA is not normally undergoing LLPS, i.e. in steady state? The authors should also make an effort to define why LLPS is measured in the way they do as this is not common knowledge in the field.

5. The authors seem to completely forget the transcriptional role of CEBPA. Does this change if they add mannitol to cells as determined by RNA-seq and CHIPseq?

6. As CEBPA-p30 can also dimerize with itself the authors should test if these dimers are undergoing LLPS

7. The authors should test if endogenous CEBPA is undergoing LLPS in a normal differentiation system. The authors should also test the impact of the S16 mutation in differentiation (NIH3T3-mediate adipocyte; see Porse et al, 2001 or 32Dcl1) systems driven by exogenous addition of CEBPA.

8. The authors findings in the MLL-AF9 mouse model are puzzling as it has been shown that CEBPA can be deleted in these cells without any apparent functional consequences. These findings suggest that the data presented in Figure 6 is not related to CEBPA function but perhaps to other CEBPs. This begs the question of whether other CEBPs can undergo LLPS. In fact, the LZ peptide would predict to influence other CEBPs as well as CEBPs are well-known to heterodimerize with each other

9. The authors claim that the effect of rapamycin is via downregulation of CEBPA-p30. However, rapamycin has MANY effects and should therefore modify their conclusion to account for other possibilities. Also, to support the data in the MLL-AF9 mouse models (Fig 6) the authors should perform westerns to assess the relative levels of CEBPA-p30 and CEBP-p42.

Minor points:

1. How many AML patients were tested in Fig1 D

2. Fig 4G. It seems unlikely that the data presented there reach statistical significance
3. Fig 6A. Which cells were used here??

Reviewer #2 (Remarks to the Author):

C/EBP α is a transcription factor associated with myeloid differentiation, and functional abnormalities such as C/EBP α -p30 mutation can lead to acute myeloid leukemia (AML). Here the article shows that homodimers of C/EBP α regulate AML cell differentiation by influencing the phase separation by promoting phosphorylation at S16. However, the p30 mutation inhibits the LLPS of C/EBP α by inhibiting homodimer binding and S16 phosphorylation levels in WT via the LZ domain. According to this property, TAT-LZ, a membrane-penetrating peptide, was designed in this research to regulate cell differentiation by inhibiting the homodimer binding of proteins and the level of phase separation.

The key mechanism of the study is relatively complete, among which the effect of homologous dimer on the phosphorylation of S16 and LLPS of protein is the innovation point. The design of interference peptide and C/EBP α -p30/C/EBP α ratio, which is associated with the treatment of acute leukemia, has certain practical significance. However, there was no direct experiment to prove the homodimers in this paper, and the function of the protein, especially its influence on the differentiation of AML cells, was not fully demonstrated. As for the key mutations in the research, such as C/EBP α -p30 and S16A/E, no relevant animal models were established for verification, which made the paper not persuasive enough to meet the publication requirements.

Major comments

1. Although it has been proved that C/EBP α -p30 mutation inhibits protein transcriptional activity by interfering with homodimerization which is a key link in this study, relevant experiments are still needed. For example, the TAT-LZ peptide affects the homodimer binding of C/EBP α , thus the binding force and expression level of the dimer can be characterized by NATIVE PAGE or MST methods, instead of assuming that proteins not bound to the peptide exist in the form of dimer.
2. As for the verification of protein function, only the transcription levels of CD11b and CD68 were used for characterization, which was not comprehensive and systematic. This study focuses on the regulation of C/EBP α on AML cell differentiation, so relevant functional experiments, especially flow cytometry experiments, are necessary, otherwise they are not convincing.
3. In this study, there were several key disease-related gene mutations, such as p30 and S16A/E, and cell transfection by constructing plasmids alone was not sufficient to simulate the disease. The corresponding transgenic model of mice is a crucial part of this project.

Minor comments

1. In line 94, "condensate" is not suitable for droplets in vitro, which is only used to describe LLPS in vivo.

Reviewer #3 (Remarks to the Author):

The article by Wang, et al., proposes that liquid-liquid phase separation (LLPS) by the C/EBP α transcription factor in the nucleus is essential for promoting myeloid differentiation. The authors show through transient overexpression in HEK293T cells and immunostaining of endogenous C/EBP α in AML patient cells and THP-1 cells that upon drug-induced differentiation, C/EBP α localizes within punctate condensates. They show both in vitro and in HEK293T cells that these condensates exhibit properties associated with LLPS (turbidity, fusion events, rapid FRAP kinetics). An AML-associated mutant of C/EBP α which lacks the N-terminal 120 residues (termed p30) is unable to undergo LLPS on its own and inhibits the LLPS of full-length C/EBP α . The authors show that this is due to heterodimerization of the truncated p30 with full-length C/EBP α via their leucine

zipper domains, thus interfering with homodimerization of the full-length protein. Further, deletion of the leucine zipper domain within the truncated p30 (C/EBP α -p30 Δ LZ construct) rescues C/EBP α LLPS, while a peptide consisting of just the LZ domain phenocopies C/EBP α -p30 and inhibits C/EBP α LLPS. The authors also identify a posttranslational modification of phospho-Serine 16 (pS16) in C/EBP α under conditions in which the protein forms robust condensates and they establish that higher pS16 levels correlate with enhanced condensation. Introducing a phospho-mimetic mutation at this site increases the number of condensates, while a phospho-null mutation decreases the number of condensates. To further support their model of C/EBP α -p30 leading to dysfunction through heterodimerization with full-length C/EBP α , the authors utilize Rapamycin, which causes degradation of the C/EBP α -p30 protein, thus tuning the ratio of C/EBP α -p30 to C/EBP α . They find that this rescues full-length C/EBP α LLPS, which also rescues expression of differentiation markers. Overall, the data in this article is of high quality and the findings are convincing, although there are several concerns that should be addressed prior to publication. The content of the article will be of interest to scientists in the transcription and biomolecular condensates fields and is the first report linking LLPS with transcriptional and developmental regulation by C/EBP α .

Major Points:

1. The authors repeatedly use mannitol-induced osmotic stress to induce condensation of C/EBP α in HEK293T cells, but never explicitly state why this treatment is used. This needs to be addressed in the text and citations showing that stress may be needed for transcription factor LLPS should be included. One example is:

Cai D, Feliciano D, Dong P, Flores E, Gruebele M, Porat-Shliom N, Sukenik S, Liu Z, Lippincott-Schwartz J. Phase separation of YAP reorganizes genome topology for long-term YAP target gene expression. *Nat Cell Biol.* 2019 Dec;21(12):1578-1589. doi: 10.1038/s41556-019-0433-z. Epub 2019 Dec 2. PMID: 31792379; PMCID: PMC8259329.

2. While the authors quantify the condensates formed by C/EBP α throughout the paper, the Methods section only states that this was done using FIJI. The authors should elaborate on the quantification method (i.e., thresholding, image processing, etc.). Additionally, the authors should show the quality of their puncta segmentation in the form of side-by-side images of the raw GFP-C/EBP α channel and the segmented puncta after quantification. These data should be shown in supplemental figures for all sets of images. This is to ensure that the condensates are being accurately counted.

3. The authors go to great lengths to show that LLPS of C/EBP α is at play during the differentiation induced by anti-leukemic drugs, but they never discuss an explicit mechanism. From their data, it is suggestive that C/EBP α forms active transcriptional condensates at genes that promote differentiation. The authors should discuss this in the Discussion section to highlight the likely mechanism underlying their findings.

4. The C/EBP α -p30 truncation results in near complete loss of C/EBP α condensates. The authors attribute this to heterodimerization of C/EBP α -p30 with full-length C/EBP α , which corresponds to the loss of one TAD1 region from the dimer. As each C/EBP α :C/EBP α -p30 heterodimer maintains one LLPS-competent TAD1 region, one would expect an attenuation of condensation but not complete loss. This suggests that TAD1 may have an additional specific role in interactions that contribute to LLPS in a dimeric context. Is there a known role of TAD1 in C/EBP α function? The authors might discuss this issue in the Discussion.

Minor Points:

1. In Fig. 1, the authors should clarify in the figure legends which cell images are showing live cells vs. fixed cells, as well as which depict exogenous over-expression vs. endogenous protein levels. This should also be done for all cell images in later figures.

2. In Fig. 2, 10% hexanediol is used on cells. The authors should specify for how long they treated cells with this concentration of hexanediol prior to imaging. This concentration of 10% hexanediol is toxic to cells and using a very short treatment time is critical.

3. Fig. 4E shows very convincing data, but the figure should be made clearer by labeling the two

axes. The "x-axis" is time after Mannitol treatment, while the "y-axis" refers to the duration of TAT-LZ peptide pre-treatment.

4. In Fig. 4g, the authors show that markers of differentiation are reduced when PMA is added in the presence of LZ fragment, compared to PMA alone. The authors should include levels of these markers in the absence of PMA as the true control, then include PMA and PMA+LZ as well.

5. In Fig. 5B-E, the authors perform western blotting with a pS16 antibody and a S16 antibody, but the specificity of these antibodies for their phospho or non-phospho targets is not discussed. The authors need to describe and demonstrate the specificity of the antibodies used and their sources.

6. In Fig. 5C, we see S16 and C/EBP α -pS16 levels decrease over the 24 hr period of PMA treatment, while the ratio of pS16/S16 increases. The authors do not comment on why C/EBP α levels decrease at all. Is there degradation of the protein? The same is not seen in Fig. 5B, where C/EBP α -S16 levels remain constant while C/EBP α -pS16 increases upon Mannitol treatment (i.e., which induces condensate formation). This discrepancy should be addressed.

7. Similarly, comparing the 24h timepoint in Fig. 5C to the 24h PMA treatment in Fig 5E shows completely different behavior of the S16 levels, which are essentially nonexistent in 5C at 24 hrs but still quite high in 5E. This discrepancy should be addressed.

8. An important observation made by the authors is that heterodimer formation of full-length C/EBP α with p30 reduces condensate formation, but they do not address the issue of the stoichiometry of the two protein forms. What mole ratio of p30 to full-length C/EBP α is needed to inhibit condensate formation? The authors should address this issue.

9. Fig. 5G shows representative images comparing condensate formation associated with expression of WT C/EBP α vs. the S16E and S16A mutants. The images for WT vs. S16A appear to be reflective of the quantification shown. Again, this is why the method of puncta quantification used by the authors should be explained and validated. The same is true of the cell images shown in Fig. 6G.

10. The qRT-PCR panel in Fig. 5H does not include any information on the cell type. This should be added to the legend or figure.

11. Figure 6A-C does not include any information on the cell type. This should be added to the legend or figure.

REVIEWER COMMENTS

Reviewer #1 (Remarks to the Author):

In the present manuscript, Wang et al demonstrate that CEBPA appear to undergo liquid-liquid phase separation (LLPS). Moreover, they claim that this is driven by formation of CEBPA-p42 homodimers, that the process is dependent on phosphorylation at S16 and that it can be inhibited by expression of CEBPA-p30.

While the demonstration of CEBPA-p42 undergoing LLPS is certainly novel the manuscript, as it stands, has a number of serious flaws and also lack demonstration of the functional relevance of their findings.

Response: We thank the reviewer for the professional comments and thoughtful suggestions, which helped improve the manuscript. We have addressed these critiques and highlighted all the changes in yellow in the revised manuscript. Please find our point-by-point responses below.

Major concerns

1. The authors seem to ignore a significant amount of recent literature on the function of CEBPA yielding a non-contemporary presentation of the field. Statements such as “...mechanism by which CEBPA functions is still unclear...” and that CEBPA-p30 should be “dominant negative” and that “CEBPA-p30 acts mainly by inhibiting the formation of CEBPA homodimers” are simply wrong.

Response: Thank you very much for your valuable and constructive comments, we apologize for the inaccurate description. We have reviewed the recent literatures on the function of C/EBP α and found that C/EBP α enhanced cell differentiation and blocked the cell cycle progression by blocking the miR-182 expression, activating *CSF3R* or *IL6R*, repressing *CEBPG* and E2F, respectively¹⁻⁷. More importantly, C/EBP α was necessary for long-term self-renewal and lineage initiation of hematopoietic stem cells⁸, and improved the curative effects of the LSD1 inhibitor in the treatment of AML⁹. In addition, its mutation has been widely used in the clinical diagnosis and treatment of AML^{10,11}. In summary, the functions of C/EBP α in various pathophysiological processes have been clarified. Further clarification of the specific molecular mechanism to achieve its functions is of great significance for the development of related drugs targeted at C/EBP α . We have revised the related descriptions in the new manuscript. (Line 18-19, 35-42)

Meanwhile, we also confirmed that, different from the “dominant negative” effect of C/EBP α -p30 in normal genotype cells which expressed C/EBP α -p42 and C/EBP α -p30 simultaneously, it works independently of C/EBP α -p42 in most patients with C/EBP α mutation. For example, C/EBP α -p30 sustained leukemic growth via the CD73/A2AR axis¹²; it alleviated immunosuppression of CD8⁺ T cells by inhibiting autophagy-associated secretion of IL-1 β in AML¹³. Thanks for your suggestion and we have modified the descriptions in this revised manuscript (Line 45-59).

References:

1. Wurm AA, et al. Disruption of the C/EBP α -miR-182 balance impairs granulocytic differentiation. *Nature communications* 8, 46 (2017).
2. Braun TP, et al. Myeloid lineage enhancers drive oncogene synergy in CEBPA/CSF3R mutant acute myeloid leukemia. *Nature communications* 10, 5455 (2019).
3. Tenen DG, Darlington GJ, Link DC, Zhang P, Iwama A, Datta MW. Upregulation of interleukin 6 and granulocyte colony-stimulating factor receptors by transcription factor CCAAT enhancer binding protein alpha (C/EBP alpha) is critical for granulopoiesis. *The Journal of Experimental Medicine*, 188 (1998).
4. Lou Y-j. CEBPA-CEBPG axis as a novel promising therapeutic target in acute myeloid leukemia. *Acta Pharmacologica Sinica* 34, 185-186 (2013).
5. Alberich-Jordà M, Wouters B, Balastik M, Shapiro-Koss C, Tenen DG. C/EBP γ deregulation results in differentiation arrest in acute myeloid leukemia. *Journal of Clinical Investigation* 122, 12 (2013).
6. Pulikkan JA, et al. Cell-cycle regulator E2F1 and microRNA-223 comprise an autoregulatory negative feedback loop in acute myeloid leukemia. *Blood* 115, 1768-1778 (2010).
7. Pulikkan JA, et al. C/EBP α regulated microRNA-34a targets E2F3 during granulopoiesis and is down-regulated in AML with CEBPA mutations. *Blood: The Journal of the American Society of Hematology*, 116 (2010).
8. Hasemann MS, et al. C/EBP α Is Required for Long-Term Self-Renewal and Lineage Priming of Hematopoietic Stem Cells and for the Maintenance of Epigenetic Configurations in Multipotent Progenitors. *PLoS genetics* 10, e1004079 (2014).
9. Cusan M, Cai SF. LSD1 inhibition exerts its antileukemic effect by recommissioning PU.1- and C/EBP α -dependent enhancers in AML. *Blood* 131, 1730-

1742 (2018).

10. Tawana K, et al. Disease evolution and outcomes in familial AML with germline CEBPA mutations. *Blood* 126, 1214 (2015).

11. Bullinger L. CEBPA mutations in AML: site matters. *Blood* 139, 6-7 (2022).

12. Jakobsen J, et al. Mutant CEBPA directly drives the expression of the targetable tumor-promoting factor CD73 in AML. *Science advances* 5, eaaw4304 (2019).

13. Wang JD, et al. Mutant C/EBP α p30 alleviates immunosuppression of CD8(+) T cells by inhibiting autophagy-associated secretion of IL-1 β in AML. *Cell Prolif* 55, e13331 (2022).

2. Also the authors should use CEBPA-p42 homodimers (instead of CEBPA homodimers) as CEBPA-p30 can dimerize with CEBPA-p42 which could, in principle be termed CEBPA homodimers.

Response: Thanks for your suggestion. We have revised the inaccurate description of C/EBP α homodimers to C/EBP α -p42 homodimers throughout the article.

3. The authors seem to think that role of CEBPA-p30 in CEBPA mutant AML is to prevent the formation of CEBPA-p42 homodimers. This is not correct as the CEBPA mutant AML is bi-allelic mutated and are either homozygous for the CEBPA-p30 allele or carries on CEBPA-p30 allele and an allele blocking CEBPA (any) dimerization. In essence this result in a situation where CEBPA-p30 homodimers are the only functional CEBPA dimers in these patients. The authors should modify the manuscript accordingly.

Response: Thank you for pointing out this issue. As you and some reports¹⁻³ mentioned, the majority of AML patients with CEBPA mutation are biallelic mutations. The selective loss of C/EBP α -p42 was found in these patients because they usually have one allele carrying the N-terminal mutation and the other carrying the C-terminal mutation. As we all know, the N-terminal mutations often expressed C/EBP α -p30 which could form C/EBP α -p30 homodimers, and the C-terminal mutations could either block dimerization of C/EBP α with itself or with other members of the CEBP family. Thus, C/EBP α -p30 homodimers are the only functional C/EBP α dimers in these patients. We are sorry for our inaccurate description of AML patients with C/EBP α mutation, and we have revised the relevant descriptions in the new manuscript according to your suggestions (Line 45-59).

References:

1. Peggy, et al. Modeling of C/EBP α Mutant Acute Myeloid Leukemia Reveals a Common Expression Signature of Committed Myeloid Leukemia-Initiating Cells. *Cancer cell* 13,299-310 (2008).
2. Wilhelmson AS, Bo TP. CCAAT enhancer binding protein alpha (CEBPA) biallelic acute myeloid leukaemia: cooperating lesions, molecular mechanisms and clinical relevance. *British Journal of Haematology* 190, 495-507 (2020).
3. Ohlsson E, Schuster MB, Hasemann M, Porse BT. The multifaceted functions of C/EBP α in normal and malignant haematopoiesis. *Leukemia* 30,767-775 (2016).
4. In order to promote LLPS the authors need to add various stressors such as mannitol, chemotherapy PEG and others. Does that mean that CEBPA is not normally undergoing LLPS, i.e. in steady state?

Response: Thank you for the suggestion. Actually, most reports¹⁻⁵ proved LLPS by using different materials, such as purified protein, overexpressed protein in pattern cells and endogenous protein under physiological conditions. In this study, we tried to add mannitol to induce the LLPS of overexpressed-C/EBP α -EGFP in HEK 293T. Then, we also found endogenous C/EBP α undergoes LLPS in PMA-treated THP-1 cells, various hormones-treated 3T3-L1 cells and G-CSF-treated 32Dcl3 cells. The three induced-differentiation cell models fully confirmed the behavioral performance of C/EBP α during the process of cell differentiation. Furthermore, endogenous C/EBP α condensates were observed in AML cells. And our peptide, which had been proved to inhibit LLPS of C/EBP α in HEK293T cell and induced-differentiation cell models, was also found to inhibit the formation of endogenous C/EBP α condensates without any stressors *in vitro* and *in vivo* (Fig. 4g, 6b and h). In summary, we believed that C/EBP α is also normally undergoing LLPS in steady state.

References:

1. Cai D, et al. Phase separation of YAP reorganizes genome topology for long-term YAP target gene expression. *Nat Cell Biol* 21, 1578-1589 (2019).
2. Gao Y, et al. Multivalent m6A motifs promote phase separation of YTHDF proteins. *Cell research* 29, 767-769 (2019).
3. Jin X, Zhou M, Chen S, Li D, Cao X. Effects of pH alterations on stress- and aging-induced protein phase separation. *Cellular and Molecular Life Sciences* 79, 380 (2022).
4. Kanaan NM, Hamel C, Grabinski T, Combs B. Liquid-liquid phase separation induces pathogenic tau conformations *in vitro*. *Nature communications* 11, 2809 (2020).

5. Wang B, et al. Condensation of SEUSS promotes hyperosmotic stress tolerance in Arabidopsis. *Nature chemical biology* 18, 1361-1369 (2022).

The authors should also make an effort to define why LLPS is measured in the way they do as this is not common knowledge in the field.

Response: Thank you for pointing out the issue. Many methods have been established to quantitatively describe the LLPS¹⁻⁶. Quantifying the number of particles in each cell is commonly used to study the difference of protein phase separation. Therefore, we counted the number of particles in each cell to characterize the phase separation. Relative descriptions have been added in the new Methods section (Line 579-591).

References:

1. Cai D, et al. Phase separation of YAP reorganizes genome topology for long-term YAP target gene expression. *Nat Cell Biol* 21, 1578-1589 (2019).
2. Liu S, Wang T, Shi Y, Bai L, Liu H. USP42 drives nuclear speckle mRNA splicing via directing dynamic phase separation to promote tumorigenesis. *Cell Death & Differentiation*.
3. Jord AA, Letort G, Eichmuller A, Chanet S, Verlhac MH. Cytoplasmic forces functionally reorganize nuclear condensates in oocytes. *Nature Communications* 13 (2021).
4. Matos C, et al. Liquid-liquid phase separation and fibrillation of the prion protein modulated by a high-affinity DNA aptamer. *FASEB journal: official publication of the Federation of American Societies for Experimental Biology* 34, 365-385 (2020).
5. Guo C, et al. ENL initiates multivalent phase separation of the super elongation complex (SEC) in controlling rapid transcriptional activation. *Science advances* 6, eaay4858 (2020).
6. Gao H, et al. Phase separation of DDX21 promotes colorectal cancer metastasis via MCM5-dependent EMT pathway. *Oncogene*, (2023).

5. The authors seem to completely forget the transcriptional role of CEBPA. Does this change if they add mannitol to cells as determined by RNA-seq and CHIPseq?

Response: We appreciate your thorough comments. In order to confirm the influence of phase separation on transcription function of C/EBP α , we performed ChIP-seq and RNA-seq for PMA treated THP-1 cells. The results of ChIP-seq showed that the binding ability of phase-separated-C/EBP α with many reported downstream target genes

*CSF3R*¹⁻², *IL6R*³ and *CEBPG*⁴ was significantly enhanced (Supplementary Fig. 2f). Meanwhile, the results of RNA-seq showed that phase-separated-C/EBP α activated the transcription of *CSF3R* and *IL6R*, but inhibited the transcription of *CEBPG* (Supplementary Fig. 2g). All the results were consistent with the previously reports and proved that the transcriptional role of C/EBP α can be activated by its LLPS. The results and relative descriptions have been added in the new version (new Supplementary Fig. 2f and g, Line 148-155).

References:

1. Friedman, Alan D. Transcriptional control of granulocyte and monocyte development. *Oncogene* 21, 3377-3390 (2007).
2. Smith LT, Hohaus S, Gonzalez DA, Dziennis SE, Tenen DG. PU.1 (Spi-1) and C/EBP alpha regulate the granulocyte colony-stimulating factor receptor promoter in myeloid cells. *Blood* 88, 1234-1247 (1996).
3. Tenen DG, Darlington GJ, Link DC, Zhang P, Iwama A, Datta MW. Upregulation of interleukin 6 and granulocyte colony-stimulating factor receptors by transcription factor CCAAT enhancer binding protein alpha (C/EBP alpha) is critical for granulopoiesis. *The Journal of Experimental Medicine*, 188 (1998).
4. Lou Y-j. CEBPA-CEBPG axis as a novel promising therapeutic target in acute myeloid leukemia. *Acta Pharmacologica Sinica* 34, 185-186 (2013).

6. As CEBPA-p30 can also dimerize with itself the authors should test if these dimers are undergoing LLPS

Response: We thank you very much for your valuable and constructive comments. As your mentioned, the C/EBP α -p30 homodimers was observed (Fig. 4a), but no condensate was found in mannitol stimulated-HEK 293T cells that were transfected with mCherry-tagged C/EBP α -p30. The results suggested that C/EBP α -p30 homodimers cannot undergo LLPS. We have added the results and relative descriptions to the revised manuscript (new Supplementary Fig. 4b, line 188-190).

7. The authors should test if endogenous CEBPA is undergoing LLPS in a normal differentiation system. The authors should also test the impact of the S16 mutation in differentiation (NIH3T3-mediate adipocyte; see Porse et al, 2001 or 32Dcl1) systems driven by exogenous addition of CEBPA.

Response: Thank you for pointing out this issue. According to your suggestion, we

detected the endogenous LLPS of C/EBP α in two normal differentiation systems¹⁻³. Firstly, when the 3T3-L1 cells differentiated into adipocytes, the endogenous C/EBP α condensates were significantly increased (Supplementary Fig. 1d-e). Then, during the differentiation of 32Dcl3 cells into granulocytes, the increase of endogenous C/EBP α condensates were also observed (Supplementary Fig. 1f-g). These results proved that endogenous C/EBP α undergoes LLPS in normal differentiation systems. We have added these results to the revised manuscript (new Supplementary Fig. 1d-g; Line 96-101).

Additionally, we overexpressed C/EBP α (WT), C/EBP α -S16E and C/EBP α -S16A in 3T3-L1 and 32Dcl3 cells to explore their effects on adipogenic or granulocyte differentiation. Compared with C/EBP α (WT), C/EBP α S16E promoted cell differentiation, while C/EBP α S16A inhibited their differentiation (Supplementary Fig. 5h-i). We have added the results to the revised manuscript (new Supplementary Fig. 5h-i; Line 266-268).

References:

1. Bo TP, Pedersen TA, Xu X, Bo L, Nerlov C. E2F repression by C/EBP α is required for adipogenesis and granulopoiesis in vivo. *Cell* 107, 247-258 (2001).
2. Cao Z, Umek RM, McKnight SL. Regulated expression of three C/EBP isoforms during adipose conversion of 3T3-L1 cells. *Genes & development* 5, 1538-1552 (1991).
3. Strom D, Cleveland J, Chellappan S, Nip J, Hiebert S. E2F-1 and E2F-3 are functionally distinct in their ability to promote myeloid cell cycle progression and block granulocyte differentiation. *Cell Growth & Differentiation: the Molecular Biology Journal of the American Association for Cancer Research* 9, 59-69 (1998).

8. The authors findings in the MLL-AF9 mouse model are puzzling as it has been shown that CEBPA can be deleted in these cells without any apparent functional consequences. These findings suggest that the data presented in Figure 6 is not related to CEBPA function but perhaps to other CEBPs.

Response: Thank you very much for your valuable and constructive comments. Porse and Tenen had used C/EBP α wild-type and knock-out MLL/AF9 mouse model to confirm that C/EBP α is required for AML initiation, but not for AML maintenance¹⁻³, which proved that C/EBP α expressed normally in MLL/AF9 mouse and played important role in AML differentiation. In our study, Ara-C and DOX were employed to

induce the differentiation of MLL-AF9 AML cells in the presence of endogenous C/EBP α (Supplementary Fig. 6c). The restored LLPS of C/EBP α was observed, which further proved the functions of C/EBP α in AML differentiation (Supplementary Fig. 6c; Line 304-306). Meanwhile, in order to prove our conclusion, we established AML xenograft mouse models by injecting NSG mice with mScarlet tagged LZ-overexpressed THP-1 cells. Expectedly, the results were identical with that in MLL-AF9 AML mouse model (Supplementary Fig. 6e-j). In summary, we believe that the LLPS of C/EBP α -p42 homodimers promotes AML differentiation. We have added the results and relative descriptions in the new version manuscript (new Supplementary Fig. 6e-j; Line 313-316).

References:

1. Ohlsson E, et al. Initiation of MLL-rearranged AML is dependent on C/EBP α . *The Journal of Experimental Medicine* 211, 5-13 (2014).
2. Levantini, et al. Hematopoietic Differentiation Is Required for Initiation of Acute Myeloid Leukemia. *Cell Stem Cell* 24, 611-623 (2015).
3. Avellino R, Delwel R. Expression and regulation of C/EBP α in normal myelopoiesis and in malignant transformation. *Blood* 129, 2083-2091 (2017).

This begs the question of whether other CEBPAs can undergo LLPS. In fact, the LZ peptide would predict to influence other CEBPs as well as CEBPs are well-known to heterodimerize with each other.

Response: We appreciate the reviewer's comments. As your suggestion, we detected the phase separation of other C/EBPs proteins and found that C/EBP δ , C/EBP ϵ and C/EBP γ could not undergo LLPS after being 10 min hyperosmotic stress stimulation (Supplementary Fig. 2e). We have added these results to the revised manuscript (Supplementary Fig. 2e; Line 141-146).

9. The authors claim that the effect of rapamycin is via downregulation of CEBPA-p30. However, rapamycin has MANY effects and should therefore modify their conclusion to account for other possibilities.

Response: Thank you for pointing out the issue. We agreed with the reviewer that rapamycin regulates AML through a variety of pathways, such as inhibiting AML cell viability via circ_0094100/miR-217/ATP1B1 axis¹ or enhancing the anti-leukemia effect as a specific inhibitor of mTOR². Therefore, we overexpressed C/EBP α -p30 in

THP-1 cells treated with PMA and rapamycin. The results of qRT-PCR and flow cytometry analysis showed that C/EBP α -p30 reversed the differentiation promoting effect of rapamycin (Fig. 6c, d and Supplementary Fig. 6b), which suggesting that rapamycin promoted differentiation mainly by down-regulation of C/EBP α -p30. We have added these results and descriptions to the revised manuscript (Fig. 6c, d and Supplementary Fig. 6b; Line 292-299).

References:

1. Cao J, Huang S, Li X. Rapamycin inhibits the progression of human acute myeloid leukemia by regulating the circ_0094100/miR-217/ATP1B1 axis. *Experimental hematology* 112-113:60-69.e62 (2022).
2. Liesveld JL, Rosell K, Lu C, Mulford D, Walker A. The mTOR Inhibitor Rapamycin Demonstrates Activity Against AML in Combination with Imatinib Mesylate and with 5-Azacytidine. *Blood* 110, 4318-4318 (2007).

Also, to support the data in the MLL-AF9 mouse models (Fig 6) the authors should perform westerns to assess the relative levels of CEBPA-p30 and CEBP-p42.

Response: We thank the reviewer for raising this point. In the revised manuscript, we evaluated the relative levels of C/EBP α -p30 and C/EBP α -p42 in MLL-AF9 mice treated with rapamycin in combination with Ara-C and Dox or not. The results showed that rapamycin decreased the relative levels of C/EBP α -p30 to C/EBP α -p42. We have updated the figures and results in the revised manuscript (new Supplementary Fig. 6c, Line 304-306).

Minor points:

1. How many AML patients were tested in Fig1 D

Response: Thanks a lot for your careful reading about our manuscript. We analyzed 6 AML patients, and at least 10 random fields were observed in each AML patient in Fig. 1d. We have added the descriptions to the revised manuscript (Line 92 and Line 846-848).

2. Fig 4G. It seems unlikely that the data presented there reach statistical significance

Response: We appreciate your detailed advice and we are sorry for the misunderstanding caused by our failure to describe the comparison. Because of the limited degree of difference, we re-synthesized the relevant primers, added the PMA

untreated group as a control and restart the qRT-PCR experiment. The results still showing significance. We provided our original data and statistics as follows for your review, so as to increase the persuasive power of our data. Meanwhile, we have updated the figures and results at the corresponding location in the revised manuscript (new Fig.4i, Line 217-221).

Original data for CD11b

	Ctrl	PMA	PMA+LZ
	Y	Y	Y
1	1.000148000	7.728638	6.105943
2	0.972799000	8.254691	6.454092
3	0.815193000	8.028966	7.236109

Statistics for PMA vs Ctrl

Statistics for PMA+LZ vs PMA

Table Analyzed	Data 1	1	Table Analyzed	Data 1
		2		
Column B	PMA	3	Column C	PMA+LZ
vs.	vs.	4	vs.	vs.
Column A	Ctrl	5	Column B	PMA
		6		
Unpaired t test		7	Unpaired t test	
P value	<0.0001	8	P value	0.0187
P value summary	****	9	P value summary	*
Significantly different (P < 0.05)?	Yes	10	Significantly different (P < 0.05)?	Yes
One- or two-tailed P value?	Two-tailed	11	One- or two-tailed P value?	Two-tailed
t, df	t=43.43 df=4	12	t, df	t=3.827 df=4
		13		
How big is the difference?		14	How big is the difference?	
Mean ± SEM of column A	0.9294 ± 0.05764, n=3	15	Mean ± SEM of column B	8.004 ± 0.1524, n=3
Mean ± SEM of column B	8.004 ± 0.1524, n=3	16	Mean ± SEM of column C	6.599 ± 0.3342, n=3
Difference between means	7.075 ± 0.1629	17	Difference between means	-1.405 ± 0.3673
95% confidence interval	6.622 to 7.527	18	95% confidence interval	-2.425 to -0.3857
R squared (eta squared)	0.9979	19	R squared (eta squared)	0.7854

Original data for CD68

	Ctrl	PMA	PMA+LZ
	Y	Y	Y
1	1.000018000	6.276783	4.547348
2	1.128984000	7.037317	4.707710
3	0.721977000	8.485651	4.890647

Statistics for PMA vs Ctrl

Statistics for PMA+LZ vs PMA

1	Table Analyzed	Data 1	1	Table Analyzed	Data 1
2			2		
3	Column B	PMA	3	Column C	PMA+LZ
4	vs.	vs.	4	vs.	vs.
5	Column A	Ctrl	5	Column B	PMA
6			6		
7	Unpaired t test		7	Unpaired t test	
8	P value	0.0007	8	P value	0.0176
9	P value summary	***	9	P value summary	*
10	Significantly different (P < 0.05)?	Yes	10	Significantly different (P < 0.05)?	Yes
11	One- or two-tailed P value?	Two-tailed	11	One- or two-tailed P value?	Two-tailed
12	t, df	t=9.586 df=4	12	t, df	t=3.893 df=4
13			13		
14	How big is the difference?		14	How big is the difference?	
15	Mean ± SEM of column A	0.9503 ± 0.1201, n=3	15	Mean ± SEM of column B	7.267 ± 0.6479, n=3
16	Mean ± SEM of column B	7.267 ± 0.6479, n=3	16	Mean ± SEM of column C	4.715 ± 0.09917, n=3
17	Difference between means	6.316 ± 0.6589	17	Difference between means	-2.551 ± 0.6554
18	95% confidence interval	4.487 to 8.146	18	95% confidence interval	-4.371 to -0.7316
19	R squared (eta squared)	0.9583	19	R squared (eta squared)	0.7912

3. Fig 6A. Which cells were used here?

Response: We appreciate the reviewer's comments and apologize for the lack of some labels. In Fig. 6a, we used THP-1 cells to confirm that rapamycin regulates the ratio of C/EBP α -p30 to C/EBP α -p42 by decreasing the endogenous C/EBP α -p30 protein level. We have added it to the revised manuscript (Line 283 and Line 943).

Review2#

C/EBP α is a transcription factor associated with myeloid differentiation, and functional abnormalities such as C/EBP α -p30 mutation can lead to acute myeloid leukemia (AML). Here the article shows that homodimers of C/EBP α regulate AML cell differentiation by influencing the phase separation by promoting phosphorylation at S16. However, the p30 mutation inhibits the LLPS of C/EBP α by inhibiting homodimer binding and S16 phosphorylation levels in WT via the LZ domain. According to this property, TAT-LZ, a membrane-penetrating peptide, was designed in this research to regulate cell differentiation by inhibiting the homodimer binding of proteins and the level of phase separation.

The key mechanism of the study is relatively complete, among which the effect of homologous dimer on the phosphorylation of S16 and LLPS of protein is the innovation point. The design of interference peptide and C/EBP α -p30/C/EBP α ratio, which is associated with the treatment of acute leukemia, has certain practical significance. However, there was no direct experiment to prove the homodimers in this paper, and the function of the protein, especially its influence on the differentiation of AML cells, was not fully demonstrated. As for the key mutations in the research, such as C/EBP α -

p30 and S16A/E, no relevant animal models were established for verification, which made the paper not persuasive enough to meet the publication requirements.

Response: We thank the reviewer for the professional comments and thoughtful suggestions to help improve the manuscript. We have addressed these critics and highlighted all the changes as yellow in the revised manuscript. Below please find out point-by-point responses:

Major comments

1. Although it has been proved that C/EBP α -p30 mutation inhibits protein transcriptional activity by interfering with homodimerization which is a key link in this study, relevant experiments are still needed. For example, the TAT-LZ peptide affects the homodimer binding of C/EBP α , thus the binding force and expression level of the dimer can be characterized by NATIVE PAGE or MST methods, instead of assuming that proteins not bound to the peptide exist in the form of dimer.

Response: We appreciate your thorough comments. Based on your suggestion, we performed experiment to confirm C/EBP α -p30 or LZ combined with C/EBP α to form heterodimers and inhibit the formation of C/EBP α homodimers. In detail, the HEK 293T cells were transfected with mCherry-tagged C/EBP α -p30 or LZ fragment together with C/EBP α constructs (Supplementary Fig. 4a). By using NATIVE PAGE, C/EBP α -p42 monomer, C/EBP α -p30 monomer, C/EBP α -p42 homodimer, C/EBP α -p30 homodimer and C/EBP α -p42: C/EBP α -p30 heterodimer were observed (Fig. 4a). More importantly, the interfering effects of C/EBP α -p30 or LZ for the formation of C/EBP α -p42 homodimer were proved. We have added the results and relative descriptions to the revised manuscript (Fig. 4a, Line 185-188).

2. As for the verification of protein function, only the transcription levels of CD11b and CD68 were used for characterization, which was not comprehensive and systematic. This study focuses on the regulation of C/EBP α on AML cell differentiation, so relevant functional experiments, especially flow cytometry experiments, are necessary, otherwise they are not convincing.

Response: Thank you for pointing out this issue. We have added the flow cytometry experiments to analyze the ratio of CD11b or CD68 positive cells. The results supported our conclusion from qRT-PCR and had been added to the revised manuscript (Fig. 4j, 5g and 6d).

3. In this study, there were several key disease-related gene mutations, such as p30 and S16A/E, and cell transfection by constructing plasmids alone was not sufficient to simulate the disease. The corresponding transgenic model of mice is a crucial part of this project.

Response: Thank you for your valuable and constructive comments. According to your suggestion, we established the MLL-AF9 AML mouse models with Control, C/EBP α (WT), C/EBP α -p30, C/EBP α -S16E and C/EBP α -S16A groups. Compared with the Control group, we found that C/EBP α -p30 significantly promoted the progression of AML, while both C/EBP α S16E and C/EBP α (WT) inhibited the progression of AML and the inhibitory effect of C/EBP α S16E was stronger than that of the WT group. All the results proved the function of C/EBP α S16 phosphorylation-mediated LLPS on AML progression and improved our models. We have added the results to the revised manuscript (Fig. 5h-i, Line 268-271).

Minor comments

1. In line 94, “condensate” is not suitable for droplets in vitro, which is only used to describe LLPS in vivo.

Response: We appreciate the reviewer’s comments. According to your suggestion and the previous study¹, we have changed the “condensate” to “micrometre-sized spheres” at the corresponding location in the revised manuscript (Line 114).

References:

1. Cai D, et al. Phase separation of YAP reorganizes genome topology for long-term YAP target gene expression. *Nat Cell Biol* 21, 1578-1589 (2019).

Reviewer #3 (Remarks to the Author):

The article by Wang, et al., proposes that liquid-liquid phase separation (LLPS) by the C/EBP α transcription factor in the nucleus is essential for promoting myeloid differentiation. The authors show through transient overexpression in HEK293T cells and immunostaining of endogenous C/EBP α in AML patient cells and THP-1 cells that upon drug-induced differentiation, C/EBP α localizes within punctate condensates. They show both in vitro and in HEK293T cells that these condensates exhibit properties associated with LLPS (turbidity, fusion events, rapid FRAP kinetics). An AML-associated mutant of C/EBP α which lacks the N-terminal 120 residues (termed p30) is

unable to undergo LLPS on its own and inhibits the LLPS of full-length C/EBP α . The authors show that this is due to heterodimerization of the truncated p30 with full-length C/EBP α via their leucine zipper domains, thus interfering with homodimerization of the full-length protein. Further, deletion of the leucine zipper domain within the truncated p30 (C/EBP α -p30 Δ LZ construct) rescues C/EBP α LLPS, while a peptide consisting of just the LZ domain phenocopies C/EBP α -p30 and inhibits C/EBP α LLPS. The authors also identify a posttranslational modification of phospho-Serine 16 (pS16) in C/EBP α under conditions in which the protein forms robust condensates and they establish that higher pS16 levels correlate with enhanced condensation. Introducing a phospho-mimetic mutation at this site increases the number of condensates, while a phospho-null mutation decreases the number of condensates. To further support their model of C/EBP α -p30 leading to dysfunction through heterodimerization with full-length C/EBP α , the authors utilize Rapamycin, which causes degradation of the C/EBP α -p30 protein, thus tuning the ratio of C/EBP α -p30 to C/EBP α . They find that this rescues full-length C/EBP α LLPS, which also rescues expression of differentiation markers. Overall, the data in this article is of high quality and the findings are convincing, although there are several concerns that should be addressed prior to publication. The content of the article will be of interest to scientists in the transcription and biomolecular condensates fields and is the first report linking LLPS with transcriptional and developmental regulation by C/EBP α .

Response: We thank the reviewer for the professional comments and thoughtful suggestions, which helped improve the manuscript. We have addressed these critiques and highlighted all the changes in yellow in the revised manuscript. Please find our point-by-point responses below.

Major Points:

1. The authors repeatedly use mannitol-induced osmotic stress to induce condensation of C/EBP α in HEK293T cells, but never explicitly state why this treatment is used. This needs to be addressed in the text and citations showing that stress may be needed for transcription factor LLPS should be included. One example is:

Cai D, Feliciano D, Dong P, Flores E, Gruebele M, Porat-Shliom N, Sukenik S, Liu Z, Lippincott-Schwartz J. Phase separation of YAP reorganizes genome topology for long-term YAP target gene expression. *Nat Cell Biol.* 2019 Dec;21(12):1578-1589. doi: 10.1038/s41556-019-0433-z. Epub 2019 Dec 2. PMID: 31792379; PMCID:

PMC8259329.

Response: We thank you very much for your valuable comments. It has been reported that LLPS is a new mechanism by which transcription factors regulate transcription processes. Some proteins, such as YAP1 and SEUSS, should be activated by LLPS to regulate gene transcription upon hyperosmotic stimulation, suggesting that the stress induced-phase separation is crucial for achieving transcription factor function¹⁻². So we assumed that transcription factor C/EBP α may undergo LLPS dependent on stress and used mannitol-induced osmotic stress to induce condensation of C/EBP α in HEK293T cells. The relative descriptions have been added in the revised manuscript (Line 127-130).

References:

1. Cai D, et al. Phase separation of YAP reorganizes genome topology for long-term YAP target gene expression. *Nat Cell Biol* 21, 1578-1589 (2019).
2. Wang B, et al. Condensation of SEUSS promotes hyperosmotic stress tolerance in Arabidopsis. *Nature chemical biology* 18, 1361-1369 (2022).

2. While the authors quantify the condensates formed by C/EBP α throughout the paper, the Methods section only states that this was done using FIJI. The authors should elaborate on the quantification method (i.e., thresholding, image processing, etc.). Additionally, the authors should show the quality of their puncta segmentation in the form of side-by-side images of the raw GFP-C/EBP α channel and the segmented puncta after quantification. These data should be shown in supplemental figures for all sets of images. This is to ensure that the condensates are being accurately counted.

Response: We thank you very much for your valuable and constructive comments. Based on your suggestion, we have updated the methods (Line 577-589) and showed the raw GFP-C/EBP α channel and the segmented puncta after quantification in the supplemental figures. Specific as follows: A set of fluorescence images from the experiment were obtained using the same microscope settings (such as resolution, optical zoom, gain, laser intensity and scan speed) to ensure consistency between the sample and the experiment. The original picture was imported into the Image J and converted into 8bit, and then adjusted the parameters to accurately identify particles. In order to accurately count the number of particles, we set different thresholds according to different experimental materials. For example, the threshold is set to 50–255 gray level for phase-separated particles produced by overexpressed-C/EBP α in HEK 293T

and THP-1 cells. For endogenous C/EBP α particles, set the thresholds of MLL-AF9 cells, AML primary cells, 3T3-L1 cells and 32Dcl3 as 90-255, 75-255, 85-255 and 60-255, which effectively eliminated the background signals in the analysis. The numbers of puncta of each cell within these parameters were evaluated. We have added the descriptions to the Method section of the revised manuscript (Line 579-591).

3. The authors go to great lengths to show that LLPS of C/EBP α is at play during the differentiation induced by anti-leukemic drugs, but they never discuss an explicit mechanism. From their data, it is suggestive that C/EBP α forms active transcriptional condensates at genes that promote differentiation. The authors should discuss this in the Discussion section to highlight the likely mechanism underlying their findings.

Response: Thanks a lot for your careful reading on our manuscript. To comprehensively describe the regulation mechanisms of C/EBP α LLPS on cell differentiation, we performed ChIP-seq (ChIP-sequencing) and RNA-seq in PMA-treated THP-1 cells, in which C/EBP α undergoes LLPS. The ChIP-seq data showed that the binding ability of phase-separated-C/EBP α with downstream target genes *CSF3R*, *IL6R* and *CEBPG* was significantly enhanced (Supplementary Fig. 2f). Simultaneously, RNA-seq results showed that C/EBP α activated the transcription of *CSF3R* and *IL6R*, but inhibited the transcription of *CEBPG* (Supplementary Fig. 2g). These targets have been reported as the direct downstream molecules of C/EBP α ¹⁻⁴, and the results proved that the LLPS of C/EBP α promoting the differentiation of AML cells by activating the transcriptional activity of its classical downstream molecules. Based on your suggestion, the added results and relative discussion have been added to the Results and Discussion sections of the revised manuscript. (Supplementary Fig. 2f-g, Line 148-154 and 365-368).

References:

1. Tenen DG, Darlington GJ, Link DC, Zhang P, Iwama A, Datta MW. Upregulation of interleukin 6 and granulocyte colony-stimulating factor receptors by transcription factor CCAAT enhancer binding protein alpha (C/EBP alpha) is critical for granulopoiesis. *The Journal of Experimental Medicine*, 188 (1998).
2. Lou Y-j. CEBPA-CEBPG axis as a novel promising therapeutic target in acute myeloid leukemia. *Acta Pharmacologica Sinica* 34, 185-186 (2013).
3. Friedman, Alan D. Transcriptional control of granulocyte and monocyte development. *Oncogene* 21, 3377-3390 (2007).

4. Smith LT, Hohaus S, Gonzalez DA, Dziennis SE, Tenen DG. PU.1 (Spi-1) and C/EBP alpha regulate the granulocyte colony-stimulating factor receptor promoter in myeloid cells. *Blood* 88, 1234-1247 (1996).

4. The C/EBP α -p30 truncation results in near complete loss of C/EBP α condensates. The authors attribute this to heterodimerization of C/EBP α -p30 with full-length C/EBP α , which corresponds to the loss of one TAD1 region from the dimer. As each C/EBP α : C/EBP α -p30 heterodimer maintains one LLPS-competent TAD1 region, one would expect an attenuation of condensation but not complete loss. This suggests that TAD1 may have an additional specific role in interactions that contribute to LLPS in a dimeric context. Is there a known role of TAD1 in C/EBP α function? The authors might discuss this issue in the Discussion.

Response: Thanks a lot for your crucial issue. Current studies have shown that TAD1 regulates transcription activation by interacting with components of RNA polymerase II pre-initiation complex, including TBP and transcription factor IIB¹. TAD1 also inhibited E2F activity and the following cell cycle operation²⁻⁴. In our study, considering that phosphorylation of C/EBP α -p42 at S16 site was essential for the LLPS, we hypothesized that S16 site in TAD1 is essential for the structural remodeling in C/EBP α -p42: C/EBP α -p30 heterodimers and has a potential significance in TAD1 functional implementation. Understanding the molecular arrangement of the C/EBP α -p42 homodimers and the specific roles of TAD1 when phase-separated C/EBP α -p42 droplets formed would be a crucial area for future study. We have added the relative discussion section in the new version of the revised manuscript (Line 335-344).

References:

1. Nerlov C, Ziff E. CCAAT/enhancer binding protein-alpha amino acid motifs with dual TBP and TFIIB binding ability co-operate to activate transcription in both yeast and mammalian cells. *The EMBO journal* 14, 4318-4328 (1995).
2. Bo TP, Pedersen TA, Xu X, Bo L, Nerlov C. E2F repression by C/EBPalpha is required for adipogenesis and granulopoiesis in vivo. *Cell* 107, 247-258 (2001).
3. Slomiany, et al. C/EBP α Inhibits Cell Growth via Direct Repression of E2F-DP-Mediated Transcription.
4. Johansen LM, et al. c-Myc is a critical target for c/EBPalpha in granulopoiesis. *Molecular and cellular biology* 21, 3789-3806 (2001).

Minor Points:

1. In Fig. 1, the authors should clarify in the figure legends which cell images are showing live cells vs. fixed cells, as well as which depict exogenous over-expression vs. endogenous protein levels. This should also be done for all cell images in later figures.

Response: We are sorry for our carelessness to clearly describe the state of cells and proteins. According to your suggestion, we have modified these relevant descriptions in all the relevant figure legends.

2. In Fig. 2, 10% hexanediol is used on cells. The authors should specify for how long they treated cells with this concentration of hexanediol prior to imaging. This concentration of 10% hexanediol is toxic to cells and using a very short treatment time is critical.

Response: Thanks a lot for your careful reading about our manuscript. We used 10% hexanediol treated cells for 2 min prior to imaging. We have added this to the revised manuscript (Line 115, 134 and 862).

3. Fig. 4E shows very convincing data, but the figure should be made clearer by labeling the two axes. The “x-axis” is time after Mannitol treatment, while the “y-axis” refers to the duration of TAT-LZ peptide pre-treatment.

Response: We thank the reviewer for the suggestions. According to your suggestion, we have labeled the two axes. The results are indicated in revised Fig. 4e.

4. In Fig. 4g, the authors show that markers of differentiation are reduced when PMA is added in the presence of LZ fragment, compared to PMA alone. The authors should include levels of these markers in the absence of PMA as the true control, then include PMA and PMA+LZ as well.

Response: We appreciate your thorough comments. Based on your suggestion, we have added the control group without PMA treating and conducted the relevant experiments again. The results showed a significant differentiation inducing effect of PMA and proved LZ fragment inhibited the differentiation process. We have updated the figures and results at the corresponding location in the revised manuscript (new Fig.4i).

5. In Fig. 5B-E, the authors perform western blotting with a pS16 antibody and a S16

antibody, but the specificity of these antibodies for their phospho or non-phospho targets is not discussed. The authors need to describe and demonstrate the specificity of the antibodies used and their sources.

Response: We appreciate the reviewer's comments and apologize for the lack of key descriptions. Actually, the p-S16 and S16 antibodies are derived from rabbits and conventional ELISA affinity testing was performed. The results showed that p-S16 antibody had up to fivefold stronger affinity for p-S16 peptide than S16 peptide. We have updated the information accordingly in the revised manuscript. (new Supplementary Fig. 5a, 232-235)

6. In Fig. 5C, we see S16 and C/EBP α -pS16 levels decrease over the 24 hr period of PMA treatment, while the ratio of pS16/S16 increases. The authors do not comment on why C/EBP α levels decrease at all. Is there degradation of the protein? The same is not seen in Fig. 5B, where C/EBP α -S16 levels remain constant while C/EBP α -pS16 increases upon Mannitol treatment (i.e., which induces condensate formation). This discrepancy should be addressed.

Response: We apologize for the confusion. In Fig. 5C, we treated THP-1 cells with PMA and the endogenous C/EBP α p-S16 and C/EBP α S16 levels were observed. Indeed, we found that the expression levels of C/EBP α -S16 decreased over the 24 hr period of PMA treatment, which is consistent with Korbinian Brand's previous work¹. Considering our results that PMA decreased the expression of C/EBP α mRNA, we believe PMA mainly inhibits the expression of C/EBP α by regulating its production. However, in Fig.5B, we used another C/EBP α -EGFP overexpressed HEK 293T cell. Considering the strong promoter mediated C/EBP α -EGFP expression in plasmids and short processing time of mannitol, we speculated that it did not affect the protein expression level of overexpressed C/EBP α , but only influenced its phosphorylation. We have provided a more detailed description at the corresponding location in the revised manuscript (Line 232-237 and 921-926).

References:

1. Gutsch R, et al. CCAAT/enhancer-binding protein beta inhibits proliferation in monocytic cells by affecting the retinoblastoma protein/E2F/cyclin E pathway but is not directly required for macrophage morphology. *The Journal of biological chemistry* 286, 22716-22729 (2011).

7. Similarly, comparing the 24h timepoint in Fig. 5C to the 24h PMA treatment in Fig 5E shows completely different behavior of the S16 levels, which are essentially nonexistent in 5C at 24 hrs but still quite high in 5E. This discrepancy should be addressed.

Response: We appreciate the reviewer's comments and apologize for our rough description. Actually, in Fig. 5C, we used THP1 to detect endogenous C/EBP α p-S16 levels. However, in Fig. 5E, we used plasmids mediated C/EBP α overexpression system to detect the effect of C/EBP α -p30 or LZ on phosphorylation of C/EBP α . We have provided a more detailed description at the corresponding location in the revised manuscript (Line 237-245).

8. An important observation made by the authors is that heterodimer formation of full-length C/EBP α with p30 reduces condensate formation, but they do not address the issue of the stoichiometry of the two protein forms. What mole ratio of p30 to full-length C/EBP α is needed to inhibit condensate formation? The authors should address this issue.

Response: Thanks a lot for your careful reading about our manuscript. After calculation, the mole ratio of C/EBP α -p30 to full-length C/EBP α is 12.6: 1, which could inhibit the condensate formation. We have added it to the figure legends in the revised manuscript (line 882).

9. Fig. 5G shows representative images comparing condensate formation associated with expression of WT C/EBP α vs. the S16E and S16A mutants. The images for WT vs. S16A appear to be reflective of the quantification shown. Again, this is why the method of puncta quantification used by the authors should be explained and validated. The same is true of the cell images shown in Fig. 6G.

Response: We thank the reviewer for the suggestions. Based on your suggestion, we have added the puncta quantification in the revised manuscript (new supplementary Fig. 4d and 5d). Specific as follows: A set of fluorescence images from the experiment were obtained using the same microscope settings (such as resolution, optical zoom, gain, laser intensity and scan speed) to ensure consistency between the sample and the experiment. The original picture was imported into the Image J and converted into 8bit, and then adjusted the parameters to accurately identify particles. In order to accurately count the number of particles, we set different thresholds according to different

experimental materials. For example, the threshold is set to 50–255 gray level for phase-separated particles produced by overexpressed-C/EBP α in HEK 293T and THP-1 cells. For endogenous C/EBP α particles, set the thresholds of MLL-AF9 cells, AML primary cells, 3T3-L1 cells and 32Dcl3 as 90-255, 75-255, 85-255 and 60-255, which effectively eliminated the background signals in the analysis. The numbers of puncta of each cell within these parameters were evaluated. We have added the descriptions to the Method section of the revised manuscript (Line 579-591).

10. The qRT-PCR panel in Fig. 5H does not include any information on the cell type. This should be added to the legend or figure.

Response: We thank the reviewer for pointing this out. In Fig.5H, we used the THP-1 cells that overexpressing C/EBP α -S16E-EGFP, C/EBP α -EGFP (WT) or C/EBP α -S16A-EGFP to study their influences on the cell differentiation. We have added the description to the legend in the revised manuscript (new Fig. 5f; line 929-931).

11. Figure 6A-C does not include any information on the cell type. This should be added to the legend or figure.

Response: We appreciate the reviewer's comments and apologize for the lack of some labels. In Fig. 6a, we used THP-1 cells to confirm that rapamycin regulates the ratio of C/EBP α -p30 to C/EBP α by decreasing the endogenous C/EBP α -p30 protein level. In Fig. 6b & 6c, we also used THP-1 cells with stable expression of mScarlet tagged LZ. We have thoroughly examined similar issues and added the description of cell type to the corresponding legend in the revised manuscript (new Fig. 6a-c, Line 283,942-951).

REVIEWER COMMENTS

Reviewer #1 (Remarks to the Author):

The manuscript has improved significantly and I applaud the authors for that. I do have one point regarding their CHIP-seq experiments, which I strongly recommend them to address. Finally, when I went through the references in the manuscript it noticed several mistakes (19 and 60 are two examples). The authors should therefore go through their references to ensure that it's correct.

In the present manuscript, Wang et al demonstrate that CEBPA appear to undergo liquid-liquid phase separation (LLPS). Moreover, they claim that this is driven by formation of CEBPA-p42 homodimers, that the process is dependent on phosphorylation at S16 and that it can be inhibited by expression of CEBPA-p30. While the demonstration of CEBPA-p42 undergoing LLPS is certainly novel the manuscript, as it stands, has a number of serious flaws and also lack demonstration of the functional relevance of their findings.

Response: We thank the reviewer for the professional comments and thoughtful suggestions, which helped improve the manuscript. We have addressed these critiques and highlighted all the changes in yellow in the revised manuscript. Please find our point-by-point responses below.

Major concerns

1. The authors seem to ignore a significant amount of recent literature on the function of CEBPA yielding a non-contemporary presentation of the field. Statements such as "...mechanism by which CEBPA functions is still unclear..." and that CEBPA-p30 should be "dominant negative" and that "CEBPA-p30 acts mainly by inhibiting the formation of CEBPA homodimers" are simply wrong.

Response: Thank you very much for your valuable and constructive comments, we apologize for the inaccurate description. We have reviewed the recent literatures on the function of C/EBP α and found that C/EBP α enhanced cell differentiation and blocked the cell cycle progression by blocking the miR-182 expression, activating CSF3R or IL6R, repressing CEBPG and E2F, respectively¹⁻⁷. More importantly, C/EBP α was necessary for long-term self-renewal and lineage initiation of hematopoietic stem cells⁸, and improved the curative effects of the LSD1 inhibitor in the treatment of AML9. In addition, its mutation has been widely used in the clinical diagnosis and treatment of AML^{10,11}. In summary, the functions of C/EBP α in various pathophysiological processes have been clarified. Further clarification of the specific molecular mechanism to achieve its functions is of great significance for the development of related drugs targeted at C/EBP α . We have revised the related descriptions in the new manuscript. (Line 18-19, 35-42)

Meanwhile, we also confirmed that, different from the "dominant negative" effect of C/EBP α -p30 in normal genotype cells which expressed C/EBP α -p42 and C/EBP α -p30 simultaneously, it works independently of C/EBP α -p42 in most patients with C/EBP α mutation. For example, C/EBP α -p30 sustained leukemic growth via the CD73/A2AR axis¹²; it alleviated immunosuppression of CD8⁺ T cells by inhibiting autophagy-associated secretion of IL-1 β in AML¹³. Thanks for your suggestion and we have modified the descriptions in this revised manuscript (Line 45-59).

References:

1. Wurm AA, et al. Disruption of the C/EBP α -miR-182 balance impairs granulocytic differentiation. *Nature communications* 8, 46 (2017).
2. Braun TP, et al. Myeloid lineage enhancers drive oncogene synergy in CEBPA/CSF3R mutant acute myeloid leukemia. *Nature communications* 10, 5455 (2019).
3. Tenen DG, Darlington GJ, Link DC, Zhang P, Iwama A, Datta MW. Upregulation of interleukin 6 and granulocyte colony-stimulating factor receptors by transcription factor CCAAT enhancer binding protein alpha (C/EBP alpha) is critical for granulopoiesis. *The Journal of Experimental Medicine*, 188 (1998).
4. Lou Y-j. CEBPA-CEBPG axis as a novel promising therapeutic target in acute

myeloid leukemia. *Acta Pharmacologica Sinica* 34, 185-186 (2013).

5. Alberich-Jordà M, Wouters B, Balastik M, Shapiro-Koss C, Tenen DG. C/EBP γ deregulation results in differentiation arrest in acute myeloid leukemia. *Journal of Clinical Investigation* 122, 12 (2013).
6. Pulikkan JA, et al. Cell-cycle regulator E2F1 and microRNA-223 comprise an autoregulatory negative feedback loop in acute myeloid leukemia. *Blood* 115, 1768-1778 (2010).
7. Pulikkan JA, et al. C/EBP α regulated microRNA-34a targets E2F3 during granulopoiesis and is down-regulated in AML with CEBPA mutations. *Blood: The Journal of the American Society of Hematology*, 116 (2010).
8. Hasemann MS, et al. C/EBP α Is Required for Long-Term Self-Renewal and Lineage Priming of Hematopoietic Stem Cells and for the Maintenance of Epigenetic Configurations in Multipotent Progenitors. *PLoS genetics* 10, e1004079 (2014).
9. Cusan M, Cai SF. LSD1 inhibition exerts its antileukemic effect by recommissioning PU.1- and C/EBP α -dependent enhancers in AML. *Blood* 131, 1730-1742 (2018).
10. Tawana K, et al. Disease evolution and outcomes in familial AML with germline CEBPA mutations. *Blood* 126, 1214 (2015).
11. Bullinger L. CEBPA mutations in AML: site matters. *Blood* 139, 6-7 (2022).
12. Jakobsen J, et al. Mutant CEBPA directly drives the expression of the targetable tumor-promoting factor CD73 in AML. *Science advances* 5, eaaw4304 (2019).
13. Wang JD, et al. Mutant C/EBP α p30 alleviates immunosuppression of CD8(+) T cells by inhibiting autophagy-associated secretion of IL-1 β in AML. *Cell Prolif* 55, e13331 (2022).

The authors have dealt with my criticism.

2. Also the authors should use CEBPA-p42 homodimers (instead of CEBPA homodimers) as CEBPA-p30 can dimerize with CEBPA-p42 which could, in principle be termed CEBPA homodimers.

Response: Thanks for your suggestion. We have revised the inaccurate description of C/EBP α homodimers to C/EBP α -p42 homodimers throughout the article.

The authors have dealt with my criticism.

3. The authors seem to think that role of CEBPA-p30 in CEBPA mutant AML is to prevent the formation of CEBPA-p42 homodimers. This is not correct as the CEBPA mutant AML is bi-allelic mutated and are either homozygous for the CEBPA-p30 allele or carries on CEBPA-p30 allele and an allele blocking CEBPA (any) dimerization. In essence this result in a situation where CEBPA-p30 homodimers are the only functional CEBPA dimers in these patients. The authors should modify the manuscript accordingly.

Response: Thank you for pointing out this issue. As you and some reports¹⁻³ mentioned, the majority of AML patients with CEBPA mutation are biallelic mutations. The selective loss of C/EBP α -p42 was found in these patients because they usually have one allele carrying the N-terminal mutation and the other carrying the C-terminal mutation. As we all know, the N-terminal mutations often expressed C/EBP α -p30 which could form C/EBP α -p30 homodimers, and the C-terminal mutations could either block dimerization of C/EBP α with itself or with other members of the CEBP family. Thus, C/EBP α -p30 homodimers are the only functional C/EBP α dimers in these patients. We are sorry for our inaccurate description of AML patients with C/EBP α mutation, and we have revised the relevant descriptions in the new manuscript according to your suggestions (Line 45-59).

References:

1. Peggy, et al. Modeling of C/EBP α Mutant Acute Myeloid Leukemia Reveals a Common Expression Signature of Committed Myeloid Leukemia-Initiating Cells. *Cancer cell* 13,299-310 (2008).
2. Wilhelmson AS, Bo TP. CCAAT enhancer binding protein alpha (CEBPA) biallelic acute myeloid leukaemia: cooperating lesions, molecular mechanisms and clinical relevance. *British Journal of Haematology* 190, 495-507 (2020).

3. Ohlsson E, Schuster MB, Hasemann M, Porse BT. The multifaceted functions of C/EBP α in normal and malignant haematopoiesis. *Leukemia* 30,767-775 (2016).

The authors have dealt with my criticism.

4. In order to promote LLPS the authors need to add various stressors such as mannitol, chemotherapy PEG and others. Does that mean that CEBPA is not normally undergoing LLPS, i.e. in steady state?

Response: Thank you for the suggestion. Actually, most reports¹⁻⁵ proved LLPS by using different materials, such as purified protein, overexpressed protein in pattern cells and endogenous protein under physiological conditions. In this study, we tried to add mannitol to induce the LLPS of overexpressed-C/EBP α -EGFP in HEK 293T. Then, we also found endogenous C/EBP α undergoes LLPS in PMA-treated THP-1 cells, various hormones-treated 3T3-L1 cells and G-CSF-treated 32Dcl3 cells. The three induced differentiation cell models fully confirmed the behavioral performance of C/EBP α during the process of cell differentiation. Furthermore, endogenous C/EBP α condensates were observed in AML cells. And our peptide, which had been proved to inhibit LLPS of C/EBP α in HEK293T cell and induced-differentiation cell models, was also found to inhibit the formation of endogenous C/EBP α condensates without any stressors in vitro and in vivo (Fig. 4g, 6b and h). In summary, we believed that C/EBP α is also normally undergoing LLPS in steady state.

References:

1. Cai D, et al. Phase separation of YAP reorganizes genome topology for long-term YAP target gene expression. *Nat Cell Biol* 21, 1578-1589 (2019).
2. Gao Y, et al. Multivalent m6A motifs promote phase separation of YTHDF proteins. *Cell research* 29, 767-769 (2019).
3. Jin X, Zhou M, Chen S, Li D, Cao X. Effects of pH alterations on stress- and aging-induced protein phase separation. *Cellular and Molecular Life Sciences* 79, 380 (2022).
4. Kanaan NM, Hamel C, Grabinski T, Combs B. Liquid-liquid phase separation induces pathogenic tau conformations in vitro. *Nature communications* 11, 2809 (2020).
5. Wang B, et al. Condensation of SEUSS promotes hyperosmotic stress tolerance in Arabidopsis. *Nature chemical biology* 18, 1361-1369 (2022).

The authors have dealt with my criticism.

The authors should also make an effort to define why LLPS is measured in the way they do as this is not common knowledge in the field.

Response: Thank you for pointing out the issue. Many methods have been established to quantitatively describe the LLPS¹⁻⁶. Quantifying the number of particles in each cell is commonly used to study the difference of protein phase separation. Therefore, we counted the number of particles in each cell to characterize the phase separation. Relative descriptions have been added in the new Methods section (Line 579-591).

References:

1. Cai D, et al. Phase separation of YAP reorganizes genome topology for long-term YAP target gene expression. *Nat Cell Biol* 21, 1578-1589 (2019).
2. Liu S, Wang T, Shi Y, Bai L, Liu H. USP42 drives nuclear speckle mRNA splicing via directing dynamic phase separation to promote tumorigenesis. *Cell Death & Differentiation*.
3. Jord AA, Letort G, Eichmüller A, Chanet S, Verlhac MH. Cytoplasmic forces functionally reorganize nuclear condensates in oocytes. *Nature Communications* 13 (2021).
4. Matos C, et al. Liquid-liquid phase separation and fibrillation of the prion protein modulated by a high-affinity DNA aptamer. *FASEB journal: official publication of the Federation of American Societies for Experimental Biology* 34, 365-385 (2020).
5. Guo C, et al. ENL initiates multivalent phase separation of the super elongation complex (SEC) in controlling rapid transcriptional activation. *Science advances* 6, eaay4858 (2020).
6. Gao H, et al. Phase separation of DDX21 promotes colorectal cancer metastasis via MCM5-dependent EMT pathway. *Oncogene*, (2023).

The authors have dealt with my criticism.

5. The authors seem to completely forget the transcriptional role of CEBPA. Does this change if they add mannitol to cells as determined by RNA-seq and CHIPseq?

Response: We appreciate your thorough comments. In order to confirm the influence of phase separation on transcription function of C/EBP α , we performed ChIP-seq and RNA-seq for PMA treated THP-1 cells. The results of ChIP-seq showed that the binding ability of phase-separated-C/EBP α with many reported downstream target genes CSF3R1-2, IL6R3 and CEBPG4 was significantly enhanced (Supplementary Fig. 2f). Meanwhile, the results of RNA-seq showed that phase-separated-C/EBP α activated the transcription of CSF3R and IL6R, but inhibited the transcription of CEBPG (Supplementary Fig. 2g). All the results were consistent with the previously reports and proved that the transcriptional role of C/EBP α can be activated by its LLPS. The results and relative descriptions have been added in the new version (new Supplementary Fig. 2f and g, Line 148-155).

References:

1. Friedman, Alan D. Transcriptional control of granulocyte and monocyte development. *Oncogene* 21, 3377-3390 (2007).
2. Smith LT, Hohaus S, Gonzalez DA, Dziennis SE, Tenen DG. PU.1 (Spi-1) and C/EBP alpha regulate the granulocyte colony-stimulating factor receptor promoter in myeloid cells. *Blood* 88, 1234-1247 (1996).
3. Tenen DG, Darlington GJ, Link DC, Zhang P, Iwama A, Datta MW. Upregulation of interleukin 6 and granulocyte colony-stimulating factor receptors by transcription factor CCAAT enhancer binding protein alpha (C/EBP alpha) is critical for granulopoiesis. *The Journal of Experimental Medicine*, 188 (1998).
4. Lou Y-j. CEBPA-CEBPG axis as a novel promising therapeutic target in acute myeloid leukemia. *Acta Pharmacologica Sinica* 34, 185-186 (2013).

The authors findings are interesting but CHIP-seq tracks from different conditions are not easily comparable as they might depend on different pulldown efficacies. Normally, this is dealt with by adding spike-in controls, which (it appears) that the authors haven't done. I would strongly recommend the authors to repeat these experiments using spike-ins. Moreover, why don't the authors try to make some more global conclusions by assessing more than three genes??

6. As CEBPA-p30 can also dimerize with itself the authors should test if these dimers are undergoing LLPS

Response: We thank you very much for your valuable and constructive comments. As your mentioned, the C/EBP α -p30 homodimers was observed (Fig. 4a), but no condensate was found in mannitol stimulated-HEK 293T cells that were transfected with mCherry-tagged C/EBP α -p30. The results suggested that C/EBP α -p30 homodimers cannot undergo LLPS. We have added the results and relative descriptions to the revised manuscript (new Supplementary Fig. 4b, line 188-190).

The authors have dealt with my criticism.

7. The authors should test if endogenous CEBPA is undergoing LLPS in a normal differentiation system. The authors should also test the impact of the S16 mutation in differentiation (NIH3T3-mediate adipocyte; see Porse et al, 2001 or 32Dcl1) systems driven by exogenous addition of CEBPA.

Response: Thank you for pointing out this issue. According to your suggestion, we detected the endogenous LLPS of C/EBP α in two normal differentiation systems1-3. Firstly, when the 3T3-L1 cells differentiated into adipocytes, the endogenous C/EBP α condensates were significantly increased (Supplementary Fig. 1d-e). Then, during the differentiation of 32Dcl3 cells into granulocytes, the increase of endogenous C/EBP α condensates were also observed (Supplementary Fig. 1f-g). These results proved that endogenous C/EBP α undergoes LLPS in normal differentiation systems. We have added these results to the revised manuscript (new Supplementary Fig. 1d-g; Line 96-101).

Additionally, we overexpressed C/EBP α (WT), C/EBP α -S16E and C/EBP α -S16A in 3T3-L1 and 32Dcl3 cells to explore their effects on adipogenic or granulocyte differentiation. Compared with C/EBP α (WT), C/EBP α S16E promoted cell differentiation, while C/EBP α S16A inhibited their differentiation (Supplementary Fig. 5h-i). We have added the results to the revised manuscript (new Supplementary Fig. 5hi; Line 266-268).

References:

1. Bo TP, Pedersen TA, Xu X, Bo L, Nerlov C. E2F repression by C/EBP α is required for adipogenesis and granulopoiesis in vivo. *Cell* 107, 247-258 (2001).
2. Cao Z, Umek RM, McKnight SL. Regulated expression of three C/EBP isoforms during adipose conversion of 3T3-L1 cells. *Genes & development* 5, 1538-1552 (1991).
3. Strom D, Cleveland J, Chellappan S, Nip J, Hiebert S. E2F-1 and E2F-3 are functionally distinct in their ability to promote myeloid cell cycle progression and block granulocyte differentiation. *Cell Growth & Differentiation: the Molecular Biology Journal of the American Association for Cancer Research* 9, 59-69 (1998).

The authors have dealt with my criticism.

8. The authors findings in the MLL-AF9 mouse model are puzzling as it has been shown that CEBPA can be deleted in these cells without any apparent functional consequences. These findings suggest that the data presented in Figure 6 is not related to CEBPA function but perhaps to other CEBPs.

Response: Thank you very much for your valuable and constructive comments. Porse and Tenen had used C/EBP α wild-type and knock-out MLL/AF9 mouse model to confirm that C/EBP α is required for AML initiation, but not for AML maintenance¹⁻³, which proved that C/EBP α expressed normally in MLL/AF9 mouse and played important role in AML differentiation. In our study, Ara-C and DOX were employed to induce the differentiation of MLL-AF9 AML cells in the present of endogenous C/EBP α (Supplementary Fig. 6c). The restored LLPS of C/EBP α was observed, which further proved the functions of C/EBP α in AML differentiation (Supplementary Fig. 6c; Line 304-306). Meanwhile, in order to prove our conclusion, we established AML xenograft mouse models by injecting NSG mice with mScarlet tagged LZoverexpressed THP-1 cells. Expectedly, the results were identical with that in MLLAF9 AML mouse model (Supplementary Fig. 6e-j). In summary, we believe that the LLPS of C/EBP α -p42 homodimers promotes AML differentiation. We have added the results and relative descriptions in the new version manuscript (new Supplementary Fig. 6e-j; Line 313-316).

References:

1. Ohlsson E, et al. Initiation of MLL-rearranged AML is dependent on C/EBP α . *The Journal of Experimental Medicine* 211, 5-13 (2014).
2. Levantini, et al. Hematopoietic Differentiation Is Required for Initiation of Acute Myeloid Leukemia. *Cell Stem Cell* 24, 611-623 (2015).
3. Avellino R, Delwel R. Expression and regulation of C/EBP α in normal myelopoiesis and in malignant transformation. *Blood* 129, 2083-2091 (2017).

The authors have dealt with my criticism.

This begs the question of whether other CEBPAs can undergo LLPS. In fact, the LZ peptide would predict to influence other CEBPs as well as CEBPs are well-known to heterodimerize with each other.

Response: We appreciate the reviewer's comments. As your suggestion, we detected the phase separation of other C/EBPs proteins and found that C/EBP δ , C/EBP ϵ and C/EBP γ could not undergo LLPS after being 10 min hyperosmotic stress stimulation (Supplementary Fig. 2e). We have added these results to the revised manuscript (Supplementary Fig. 2e; Line 141-146).

The authors have dealt with my criticism.

9. The authors claim that the effect of rapamycin is via downregulation of CEBPA-p30.

However, rapamycin has MANY effects and should therefore modify their conclusion to account for other possibilities.

Response: Thank you for pointing out the issue. We agreed with the reviewer that rapamycin regulates AML through a variety of pathways, such as inhibiting AML cell viability via circ_0094100/miR-217/ATP1B1 axis1 or enhancing the anti-leukemia effect as a specific inhibitor of mTOR2. Therefore, we overexpressed C/EBP α -p30 in THP-1 cells treated with PMA and rapamycin. The results of qRT-PCR and flow cytometry analysis showed that C/EBP α -p30 reversed the differentiation promoting effect of rapamycin (Fig. 6c, d and Supplementary Fig. 6b), which suggesting that rapamycin promoted differentiation mainly by down-regulation of C/EBP α -p30. We have added these results and descriptions to the revised manuscript (Fig. 6c, d and Supplementary Fig. 6b; Line 292-299).

References:

1. Cao J, Huang S, Li X. Rapamycin inhibits the progression of human acute myeloid leukemia by regulating the circ_0094100/miR-217/ATP1B1 axis. *Experimental hematology* 112-113:60-69.e62 (2022).

2. Liesveld JL, Rosell K, Lu C, Mulford D, Walker A. The mTOR Inhibitor Rapamycin Demonstrates Activity Against AML in Combination with Imatinib Mesylate and with 5-Azacytidine. *Blood* 110, 4318-4318 (2007).

Also, to support the data in the MLL-AF9 mouse models (Fig 6) the authors should perform westerns to assess the relative levels of CEBPA-p30 and CEBP-p42.

Response: We thank the reviewer for raising this point. In the revised manuscript, we evaluated the relative levels of C/EBP α -p30 and C/EBP α -p42 in MLL-AF9 mice treated with rapamycin in combination with Ara-C and Dox or not. The results showed that rapamycin decreased the relative levels of C/EBP α -p30 to C/EBP α -p42. We have updated the figures and results in the revised manuscript (new Supplementary Fig. 6c, Line 304-306).

The authors have dealt with my criticism.

Minor points:

1. How many AML patients were tested in Fig1 D

Response: Thanks a lot for your careful reading about our manuscript. We analyzed 6 AML patients, and at least 10 random fields were observed in each AML patient in Fig. 1d. We have added the descriptions to the revised manuscript (Line 92 and Line 846-848).

The authors have dealt with my criticism.

2. Fig 4G. It seems unlikely that the data presented there reach statistical significance

Response: We appreciate your detailed advice and we are sorry for the misunderstanding caused by our failure to describe the comparison. Because of the limited degree of difference, we re-synthesized the relevant primers, added the PMA untreated group as a control and restart the qRT-PCR experiment. The results still showing significance. We provided our original data and statistics as follows for your review, so as to increase the persuasive power of our data. Meanwhile, we have updated the figures and results at the corresponding location in the revised manuscript (new Fig.4i, Line 217-221).

Original data for CD11b

Statistics for PMA vs Ctrl Statistics for PMA+LZ vs PMA

Original data for CD68

Statistics for PMA vs Ctrl Statistics for PMA+LZ vs PMA

The authors have dealt with my criticism.

3. Fig 6A. Which cells were used here?

Response: We appreciate the reviewer's comments and apologize for the lack of some

labels. In Fig. 6a, we used THP-1 cells to confirm that rapamycin regulates the ratio of C/EBP α -p30 to C/EBP α -p42 by decreasing the endogenous C/EBP α -p30 protein level. We have added it to the revised manuscript (Line 283 and Line 943).

The authors have dealt with my criticism.

Reviewer #2 (Remarks to the Author):

I have no more questions.

Reviewer #3 (Remarks to the Author):

The revised manuscript is significantly improved over the original and the authors have appropriately addressed most of this reviewer's previous concerns. However, there are a few points that need to be addressed before final consideration of the manuscript. In addition, there are several sentences that would benefit from rephrasing as the current wording may lead to misinterpretation or the statements made are not fully supported by the presented data. Suggested rephrasing of several sentences is provided at the end.

Major Point:

1. Pertaining to the response for the previous concern:

"8. An important observation made by the authors is that heterodimer formation of full-length C/EBP α with p30 reduces condensate formation, but they do not address the issue of the stoichiometry of the two protein forms. What mole ratio of p30 to full-length C/EBP α is needed to inhibit condensate formation? The authors should address this issue."

"Response: Thanks a lot for your careful reading about our manuscript. After calculation, the mole ratio of C/EBP α -p30 to full-length C/EBP α is 12.6: 1, which could inhibit the condensate formation. We have added it to the figure legends in the revised manuscript (line 882)."

While it is appreciated that the authors attempted to address the concern, it is unclear how they did so. There are no details in the methods section or anywhere else in the manuscript to describe how the 12.6:1 ratio was calculated from Fig. 3G. To clarify the original comment, it is unclear what molar ratio of p30 to p42 is needed to inhibit condensate formation. To calculate this using microscopy images like those in Fig. 3G, the authors would need to establish standard curves for the relationships between EGFP and mCherry fluorescence intensity and molar concentration. This can be done using purified EGFP and mCherry. Having these curves would allow independent determination of the nuclear concentrations of C/EBP α -EGFP and mCherry-C/EBP α -p30, and to then determine the mole ratio of the two. The authors need to explain how they determined the mole ratio of 12.6:1 and revise their approach if not done as suggested above.

Minors Points:

1. In Fig. 1C, the flow cytometry graphics should also be labeled with either Control or D/A.

2. In Fig. 2D and 2H, the provided FRAP curves should include a pre-bleach or time = 0 sec relative intensity data point. Presumably, it is 1.0 in this case, but the authors should include this point in the graph, as is customary in the field.

3. In Fig. 3G, while it is clear that the p30- Δ LZ construct fails to interfere with condensate formation by p42 in the presence of mannitol, the authors do not comment on the fact that mCherry-p30 Δ LZ localized within condensates together with p42 (e.g., the lower right panel in Fig.

3G shows mostly overlapping (yellow) condensates). As the IP in Fig. 3F shows no direct interaction between these two proteins, this result is puzzling. Do the authors have an explanation for this? Does the remaining DNA binding portion of the coiled-coil mediate binding to the same DNA sites as p42 and p30? Have the authors determined the localization of cells transfected with mCherry-p30- Δ LZ alone? If not, this experiment is suggested to possibly provide an explanation for this somewhat contradictory result.

4. Fig. 4A includes a native PAGE Western blot. The authors should label on the actual figure what is being probed in each panel (top: anti-C/EBPA, middle: anti-mCherry). Were the two large blots made from the same membrane? The aspect ratios appear to be different, which is distracting when analyzing the two blots. The authors may consider remaking the figure for clarity.

5. In line 368, the authors write: "The results could improve the efficacy of drugs in AML cells and provide new insights for the treatment of the disease". Could the authors please clarify what is meant by the first phrase ("improving the efficacy of drugs in AML cells")? Please clarify what is meant by rephrasing the sentence.

Sentence rephrasing suggestions:

1. Line 143-146:

"To detect the LLPS of C/EBP family, we studied the phase separation of C/EBPA, C/EBP ϵ and C/EBP γ and no their LLPS was observed after the cells being stimulated for 10 min by hyperosmotic stress (Supplementary Fig. 2e)."

Can be changed to:

"To probe the LLPS potential of other C/EBP family members, we expressed C/EBP δ , C/EBP ϵ and C/EBP γ in HEK 293T cells but did not observe puncta formation after 10 minutes of hyperosmotic stress (Supplementary Fig. 2e)."

2. Line 155:

"The transcription function of C/EBPa can be activated by its LLPS."

Can be changed to:

"The transcriptional activity of C/EBPa correlates with its ability to undergo phase separation."

3. Line 365-368:

"Meanwhile, our results from CHIP-seq and RNA-seq proved that the LLPS of C/EBPa promoting the differentiation of AML cells by activating the transcriptional activity of its classical downstream molecules including CSF3R, IL6R and CEBPG"

Can be changed to:

"Meanwhile, our results from CHIP-seq and RNA-seq suggest that phase separation by C/EBPa promoted differentiation of AML cells through transcriptional regulation of several of its classical downstream targets, including CSF3R, IL6R and CEBPG."

REVIEWER COMMENTS

Reviewer #1 (Remarks to the Author):

The manuscript has improved significantly and I applaud the authors for that.

Response: We thank the reviewer for your approval of the revised manuscript.

I do have one point regarding their CHIP-seq experiments, which I strongly recommend them to address.

Response: Thank you very much for your professional opinions and thoughtful suggestions again, which is helpful to improve the manuscript. For the CHIP-seq experiment, we have addressed this issue in the following “Question 5” and highlighted all the changes in yellow in the newly version manuscript (new Supplementary Fig. 2f-h, Line 150-157; Line 393-399 and Line 473-510).

Finally, when I went through the references in the references in the manuscript it noticed several mistakes (19 and 60 are two examples). The authors should therefore go through their references to ensure that it's correct.

Response: We appreciate the reviewer's comments and apologize for our carelessness. Based on your suggestion, we have checked and revised all the references in the newly manuscript.

5. The authors findings are interesting but CHIP-seq tracks from different conditions are not easily comparable as they might depend on different pulldown efficacies. Normally, this is dealt with by adding spike-in controls, which (it appears) that the authors haven't done. I would strongly recommend the authors to repeat these experiments using spike-ins. Moreover, why don't the authors try to make some more global conclusions by assessing more than three genes??

Response: Thank you very much for your valuable and constructive comments. According to your suggestion, we have added the Drosophila genome as a spike-in controls to deal with the issue of different pull-down efficiency. Meanwhile, we also analyzed the data from a macro perspective to make some more global conclusions. The results of GSEA analysis of CHIP-seq and RNA-seq showed that the phase separated-C/EBP α in PMA treated THP-1 cells activated many differentiation-related pathways, including developmental growth involved in morphogenesis, myeloid cell

differentiation, developmental maturation and macrophage activation. The results and relative descriptions have been added in the new version manuscript (new Supplementary Fig. 2f-h, Line 150-157; Line 393-399 and Line 473-510).

Reviewer #2 (Remarks to the Author):

I have no more questions.

Reviewer #3 (Remarks to the Author):

The revised manuscript is significantly improved over the original and the authors have appropriately addressed most of this reviewer's previous concerns. However, there are a few points that need to be addressed before final consideration of the manuscript. In addition, there are several sentences that would benefit from rephrasing as the current wording may lead to misinterpretation or the statements made are not fully supported by the presented data. Suggested rephrasing of several sentences is provided at the end.

Response: We thank the reviewers for your approval of the revised manuscript and put forward professional opinions and thoughtful suggestions again, which is helpful to improve the manuscript. We have addressed these critics and highlighted all the changes as yellow in the newly version manuscript. Below please find out point-by-point responses:

Major Point:

1. Pertaining to the response for the previous concern:

“8. An important observation made by the authors is that heterodimer formation of full-length C/EBP α with p30 reduces condensate formation, but they do not address the issue of the stoichiometry of the two protein forms. What mole ratio of p30 to full-length C/EBP α is needed to inhibit condensate formation? The authors should address this issue.”

“ Response: Thanks a lot for your careful reading about our manuscript. After

calculation, the mole ratio of C/EBP α -p30 to full-length C/EBP α is 12.6: 1, which could inhibit the condensate formation. We have added it to the figure legends in the revised manuscript (line 882).”

While it is appreciated that the authors attempted to address the concern, it is unclear how they did so. There are no details in the methods section or anywhere else in the manuscript to describe how the 12.6:1 ratio was calculated from Fig. 3G. To clarify the original comment, it is unclear what molar ratio of p30 to p42 is needed to inhibit condensate formation. To calculate this using microscopy images like those in Fig. 3G, the authors would need to establish standard curves for the relationships between EGFP and mCherry fluorescence intensity and molar concentration. This can be done using purified EGFP and mCherry. Having these curves would allow independent determination of the nuclear concentrations of C/EBP α -EGFP and mCherry-C/EBP α -p30, and to then determine the mole ratio of the two. The authors need to explain how they determined the mole ratio of 12.6:1 and revise their approach if not done as suggested above.

Response: Thank you very much for your valuable and constructive comments. According to your suggestion, we have added a concentration gradient experiment in mannitol stimulated-HEK 293T cells. The results from the mass ratio of C/EBP α -p30 construct to full-length C/EBP α construct as 1:1, 3:1, 9:1 and 18:1 (mole ratio: 1.4:1, 4.2:1, 12.6:1 and 25.2:1) showed that the number of condensates in 1:1 group was obviously more than that in 3:1 group (Supplementary Fig. 3d-e). However, no condensate was observed in 9:1 or 18:1 group (Supplementary Fig. 3d-e). Meanwhile, similar results have also been obtained in experiment by using purified mCherry-tagged C/EBP α -p30 and C/EBP α -EGFP proteins (Supplementary Fig. 3f). Collectively, as the mass ratio of C/EBP α -p30 to C/EBP α increases, the condensates gradually decreased until they disappeared. The mass ratio of C/EBP α -p30 construct to full-length C/EBP α construct 9:1 (mole ratio: 12.6:1) is the lowest mass ratio for C/EBP α -p30 to suppress C/EBP α LLPS, and was selected in subsequent experiments. The results and relative descriptions have been added in the new version (new Supplementary Fig. 3d-f and Line 174-187; Supplementary Line 29-39).

Minors Points:

1. In Fig. 1C, the flow cytometry graphics should also be labeled with either Control or

D/A.

Response: We appreciate the reviewer's comments. Based on your suggestion, we have added the labels in the newly revised manuscript (Fig. 1c).

2. In Fig. 2D and 2H, the provided FRAP curves should include a pre-bleach or time = 0 sec relative intensity data point. Presumably, it is 1.0 in this case, but the authors should include this point in the graph, as is customary in the field.

Response: We appreciate the reviewer's comments. According to your suggestion and the previous study^[1-3], we have added the relative intensity of pre-bleaching (normalized to 1) in the revised manuscript (Fig.2d and 2h).

References:

1. Fang, X., et al., Arabidopsis FLL2 promotes liquid-liquid phase separation of polyadenylation complexes. *Nature* **569**, 265-269(2019).
2. White, M., et al., C9orf72 Poly(PR) Dipeptide Repeats Disturb Biomolecular Phase Separation and Disrupt Nucleolar Function. *Molecular cell* **74**, 713-728(2019).
3. Kanaan, N., et al., Liquid-liquid phase separation induces pathogenic tau conformations in vitro. *Nature communications* **11**, 2809(2020).

3. In Fig. 3G, while it is clear that the p30- Δ LZ construct fails to interfere with condensate formation by p42 in the presence of mannitol, the authors do not comment on the fact that mCherry-p30 Δ LZ localized within condensates together with p42 (e.g., the lower right panel in Fig. 3G shows mostly overlapping (yellow) condensates). As the IP in Fig. 3F shows no direct interaction between these two proteins, this result is puzzling. Do the authors have an explanation for this? Does the remaining DNA binding portion of the coiled-coil mediate binding to the same DNA sites as p42 and p30? Have the authors determined the localization of cells transfected with mCherry-p30- Δ LZ alone? If not, this experiment is suggested to possibly provide an explanation for this somewhat contradictory result.

Response: We thank you very much for your valuable and constructive comments. According to your suggestion, we observed the LLPS of mCherry-tagged C/EBP α -p30 Δ LZ in mannitol stimulated-HEK 293T cells and no condensate was found, which suggested that C/EBP α -p30 Δ LZ cannot undergo LLPS (Supplementary Fig. 3g). Meanwhile, considering that C/EBP α -p30 Δ LZ and C/EBP α had no direct interaction

and some reports mentioned that they could bind to the same gene promoters ^[1,2], we speculate that the LLPS of C/EBP α changed the distribution and functional states of its binding chromosomes, which led to the redistribution of C/EBP α -p30 Δ LZ which had a similar chromosome binding ability as wild type C/EBP α . We have added the relative discussion in the revised manuscript (Supplementary Fig. 3g, Line 190-198 and 346-352).

References:

1. Paz-Priel, I., et al., C/EBP α or C/EBP α oncoproteins regulate the intrinsic and extrinsic apoptotic pathways by direct interaction with NF- κ B p50 bound to the bcl-2 and FLIP gene promoters. *Leukemia* **23**, 365–374 (2009).
2. Gentle, I.E., et al., The AML-associated K313 mutation enhances C/EBP α activity by leading to C/EBP α overexpression. *Cell Death & Disease* **12**, 675(2021).

4. Fig. 4A includes a native PAGE Western blot. The authors should label on the actual figure what is being probed in each panel (top: anti-C/EBP α , middle: anti-mCherry). Were the two large blots made from the same membrane? The aspect ratios appear to be different, which is distracting when analyzing the two blots. The authors may consider remaking the figure for clarity.

Response: We appreciate the reviewer's comments. Based on your suggestion, we repeated the experiment by incubating the same membrane with different antibodies. Consistent with the previous results, C/EBP α -p42 monomer, C/EBP α -p30 monomer, C/EBP α -p42 homodimer, C/EBP α -p30 homodimer and C/EBP α -p42: C/EBP α -p30 heterodimer were observed. More importantly, the interfering effects of C/EBP α -p30 or LZ for the formation of C/EBP α -p42 homodimer were confirmed. We have provided more detailed and accurate annotations for the images in the revised manuscript (Fig. 4a).

5. In line 368, the authors write: "The results could improve the efficacy of drugs in AML cells and provide new insights for the treatment of the disease".

Could the authors please clarify what is meant by the first phrase ("improving the efficacy of drugs in AML cells")? Please clarify what is meant by rephrasing the sentence.

Response: We appreciate the reviewer's comments and apologize for the conclusion that is not accurate. We have modified these relevant descriptions in the revised

manuscript to “The results reveal a new mechanism of C/EBP α regulating AML cell differentiation and provide new insights for the treatment of the AML.” (Line 397-399), which provides a clearer explanation of our study.

Sentence rephrasing suggestions:

1. Line 143-146:

“To detect the LLPS of C/EBP family, we studied the phase separation of C/EBP α , C/EBP ϵ and C/EBP γ and no their LLPS was observed after the cells being stimulated for 10 min by hyperosmotic stress (Supplementary Fig. 2e).”

Can be changed to:

“To probe the LLPS potential of other C/EBP family members, we expressed C/EBP α , C/EBP ϵ and C/EBP γ in HEK 293T cells but did not observe puncta formation after 10 minutes of hyperosmotic stress (Supplementary Fig. 2e).”

2. Line 155:

“The transcription function of C/EBP α can be activated by its LLPS.”

Can be changed to:

“The transcriptional activity of C/EBP α correlates with its ability to undergo phase separation.”

3. Line 365-368:

“Meanwhile, our results from CHIP-seq and RNA-seq proved that the LLPS of C/EBP α promoting the differentiation of AML cells by activating the transcriptional activity of its classical downstream molecules including CSF3R, IL6R and CEBPG”

Can be changed to:

“Meanwhile, our results from CHIP-seq and RNA-seq suggest that phase separation by C/EBP α promoted differentiation of AML cells through transcriptional regulation of several of its classical downstream targets, including CSF3R, IL6R and CEBPG.”

Response: Thanks a lot for your careful reading about our manuscript. According to your suggestion, we have updated and modified all these sentences in the revised manuscript (Line 143-146; Line 156-157; Line 393-397).

REVIEWERS' COMMENTS

Reviewer #1 (Remarks to the Author):

I am happy with the changes made by the authors, but I encourage them to rephrase the new section (lines 393-399). First, I assume that "presents condensates" should be "prevents condensates"?. Also, it is not correct to state that there is no direct binding between the p42 and p30. Both have the CEBPA dimerization domain so they would be expected to form heterodimers!

Reviewer #3 (Remarks to the Author):

The authors have addressed this reviewer's suggestions on the revised manuscript. The work is now suitable for publication.

REVIEWERS' COMMENTS

Reviewer #1 (Remarks to the Author):

I am happy with the changes made by the authors, but I encourage them to rephrase the new section (lines 393-399). First, I assume that “presents condensates” should be “prevents condensates”?.

Response: Thanks a lot for your careful reading about our manuscript. According to your suggestion, we have updated and modified the sentence in the revised manuscript (Line 346-347).

Also, it is not correct to state that there is no direct binding between the p42 and p30. Both have the CEBPA dimerization domain so they would be expected to form heterodimers!

Response: We appreciate the reviewer’s comments and apologize for the confused description. We have modified these relevant descriptions in the revised manuscript (Line 347-349).